# GENERATIVE MODELING FOR MULTI-TASK VISUAL LEARNING

## ABSTRACT

Generative modeling has recently shown great promise in computer vision, but it has mostly focused on synthesizing visually realistic images. In this paper, motivated by multi-task learning of shareable feature representations, we consider a novel problem of learning a shared generative model that is useful across various visual perception tasks. Correspondingly, we propose a general multi-task oriented generative modeling (MGM) framework, by coupling a discriminative multi-task network with a generative network. While it is challenging to synthesize both RGB images and pixel-level annotations in multi-task scenarios, our framework enables us to use synthesized images paired with only weak annotations (*i.e.*, image-level scene labels) to facilitate multiple visual tasks. Experimental evaluation on challenging multi-task benchmarks, including NYUv2 and Taskonomy, demonstrates that our MGM framework improves the performance of all the tasks by large margins, especially in the low-data regimes, and our model consistently outperforms state-of-the-art multi-task approaches.

## 1 INTRODUCTION

Seeing with the mind's eye — creating internal images of objects and scenes not actually present to the senses, is perhaps one of the hallmarks in human cognition (Pelaprat & Cole, 2011). For humans, this visual imagination integrates learning experience and facilitates learning by solving different problems (Egan, 2014; Pearson, 2019; Pelaprat & Cole, 2011; Egan, 1989). Inspired by such ability, there has been increasing interest in building generative models that can synthesize images (Goodfellow et al., 2014). Yet, most of the effort has focused on generating visually realistic images (Brock et al., 2018; Zhang et al., 2019), which are still far from useful for machine perception tasks (Shmelkov et al., 2018; Borji, 2019; Wu et al., 2016). Even though recent work has started improving the "usefulness" of synthesized images, this line of investigation is often limited to a single specific task (Nguyen-Phuoc et al., 2018; Sitzmann et al., 2019; Zhu et al., 2018; Souly et al., 2017). Could we guide generative models to benefit *multiple* visual tasks?

While similar spirits of shared feature representations have been widely studied as multi-task learning or meta-learning (Finn et al., 2017; Zamir et al., 2018), here we take a different perspective — *learning a shared generative model across various tasks* (as illustrated in Figure 1). Leveraging multiple tasks allows us to capture the underlying image generation mechanism for more comprehensive object and scene understanding than being done within individual tasks. Taking simultaneous semantic segmentation, depth estimation, and surface normal prediction as an example (Figure 1), successful generative modeling requires understanding not only the semantics but also the 3D geometric structure and physical property of the input image. Meanwhile, a learned generative model facilitates the flow of knowledge across tasks, so that they benefit one another. For instance, the synthesized images provide meaningful variations in existing images and could work as additional training data to build better task-specific models. These variations are especially critical when the data is limited.

This paper thus explores *multi-task oriented generative modeling* (MGM), by coupling a discriminative multi-task network with a generative network. To make them cooperate with each other, a straightforward solution would be to synthesize both RGB images and corresponding *pixel-level annotations* (*e.g.*, pixel-wise class labels for semantic segmentation and depth map for normal estimation). In the single task scenario, existing work trains a separate generative model to synthesize paired pixel-level labeled data (Sandfort et al., 2019; Choi et al., 2019) and produce an augmented set. However, these models are still highly task-dependant, and extending them to multi-task scenarios becomes difficult. A natural question then is: Do we actually need to synthesize paired image and multi-annotation data to be useful for multi-task visual learning?

Figure 1: **Left:** Traditional multi-task learning framework (that learns shared feature representations) **vs. Right:** our proposed multi-task oriented generative modeling (that learns a shared generative model across various visual perception tasks)

Our MGM addresses this question by proposing a *general* framework that uses synthesized images paired with *only weak annotations* (*i.e.*, image-level scene labels) to facilitate multiple visual tasks. Our key insight is to introduce *auxiliary discriminative tasks* that (i) only require image-level annotation or no annotation, and (ii) correlate with the original multiple tasks of interest. To this end, as additional components of the discriminative multi-task network, we introduce a *refinement* network and a *self-supervision* network that satisfies these properties. Through joint training, the discriminative network *explicitly* guides the image synthesis process. The generative network also contributes to further refining the shared feature representation. Meanwhile, the synthesized images of the generative network are used as additional training data for the discriminative network.

In more detail, the refinement network performs scene classification on the basis of the multi-task network predictions, which requires only scene labels. The self-supervision network can be operationalized on both real and synthesized images without reliance on annotations. With these two modules, our MGM is able to learn from both (pixel-wise) fully-annotated real images and synthesized (image-level) weakly labeled images. We instantiate MGM with the state-of-the-art encoder-decoder based multi-task network (Zamir et al., 2018), self-attention GAN (Zhang et al., 2019), and contrastive learning-based self-supervision network (Chen et al., 2020). Note that our framework is *agnostic to the choice of these model components*.

We evaluate our approach on standard multi-task benchmarks, including the NYUv2 (Nathan Silberman & Fergus, 2012) and Taskonomy (Zamir et al., 2018) datasets. Consistent with the previous works (Sun et al., 2019; Standley et al., 2020), we focus on three tasks of great practical importance: semantic segmentation, depth estimation, and normal prediction. The evaluation shows that: **(1)** our MGM consistently outperforms state-of-the-art multi-task approaches by large margins, especially in the low-data regime. **(2)** With the increasing of generated samples, the performance of MGM consistently improves and it also almost reaches the *performance upper-bound* that trains with weakly annotated *real* images. **(3)** Our model can be extended to more visual tasks.

## 2 PILOT STUDY

This pilot study provides an initial experimentation, which shows the importance of our proposed problem (multi-task oriented generative modeling) and motivates the development of our method. Specifically, we show that *directly using images synthesized by an off-the-shelf generative model that is trained with the realistic objective is not helpful for downstream pixel-level perception tasks*.

**Experimental Design:** For ease of analysis, here we focus on a single task – semantic segmentation, and use the Tiny-Taskonomy dataset (Zamir et al., 2018). The dataset split and evaluation metric are the same as our main experiments (See Section 4 for details). We train a self-attention GAN (Zhang et al., 2019) on Tiny-Taskonomy, and use it to generate the same number of synthesized images as the real images to augment the training set.

**How to generate pixel-level annotations?** One remaining question is how to generate pixel-level annotations for these synthesized images produced by the self-attention GAN. While prior work has explored synthesizing both images and their pixel-level annotations for some specific tasks (Sandfort et al., 2019; Choi et al., 2019), these annotations are still not reliable. For ease of analysis, in this study, we factor out the effect of annotations and assume that we have an *oracle annotator*. We use the annotator from Taskonomy (Zamir et al., 2018), which is a state-of-the-art large fully supervised semantic segmentation network. In fact, the ground-truth of semantic segmentation on Taskonomy is produced as the output of this network rather than labeled by humans. By doing so, we ensure that the annotations of the synthesized images are "accurate".

**Comparisons: Single-Task (ST)** model is our baseline which follows the architecture of Taskonomy single task network. ST is trained on real images only. $ST_G$ is the ST model trained on the augmented set. Table 1 reports the results of these two models, and $ST_G$ is worse than **ST**.

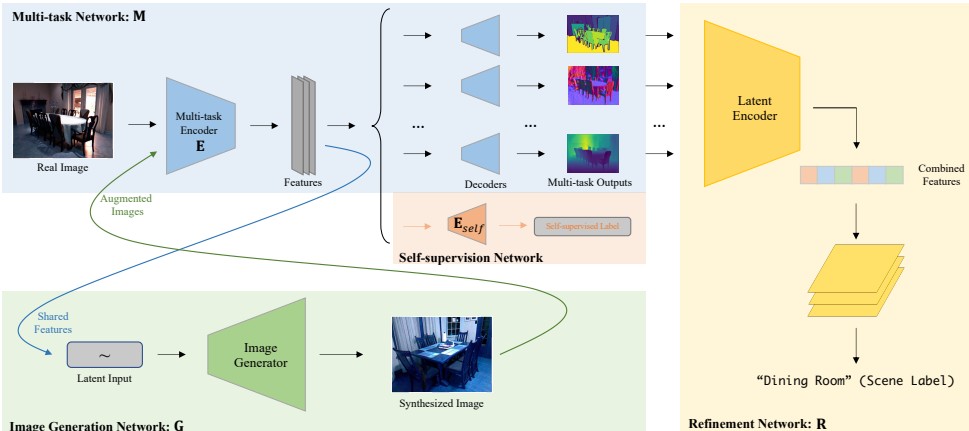

Figure 2: Architecture of our proposed multi-task oriented generative modeling (MGM) framework. There are four main components in the framework: Multi-task network to address the target multiple pixel-level prediction tasks; self-supervision network to facilitate representation learning using images without any annotation; refinement network to perform scene classification using weak annotation; image generation network to synthesize useful images that benefit multiple tasks.

| Model | ST | $\text{ST}_\text{G}$ |
|---|---|---|
| mLoss ($\downarrow$) | 0.111 | 0.148 |

Table 1: Pilot experiment for semantic segmentation on the Tiny-Taskonomy dataset. Directly using images synthesized by an off-the-shelf generative model (self-attention GAN) may hurt the performance on the downstream task. ST: single-task semantic segmentation model trained on real images only; $\text{ST}_\text{G}$: the same model trained on both real and synthesized images.

**Does synthesized images help downstream tasks?** From Table 1, the answer is **NO**. Even though the images are synthesized by one of the state-of-the-art generative models and are labeled by the oracle annotator, they still cannot benefit the downstream task. This is probably because these images are synthesized off the shelf *without* "knowing" the downstream task. *Our key insight* then is that we need to explicitly use the downstream task objective to guide the image synthesis process. Moreover, here we focused on a single task and assumed that we had the oracle annotations. However, an oracle annotator is difficult to obtain in practice, especially for multiple tasks. Also, existing work cannot synthesize paired images and pixel-level annotations for multiple tasks (Sandfort et al., 2019; Choi et al., 2019). To overcome these challenges, in what follows we demonstrate how to facilitate multiple visual tasks with synthesize images that only need image-level scene labels.

## 3 METHOD

We propose multi-task oriented generative modeling (MGM) to leverage generative networks for multi-task visual learning, as summarized in Figure 2. In this section, we first formalize the novel problem setting of MGM. Then, we explain the general framework and an instantiation of the MGM model with state-of-the-art multi-task learning and image generation approaches. Finally, we discuss the detailed training strategy for the framework.

### 3.1 PROBLEM SETTING

**Multi-task discriminative learning:** Given a set of $n$ visual tasks $\mathcal{T} = \{T_1, T_2, \cdots, T_n\}$, we aim to learn a discriminative multi-task model $\mathbf{M}$ that is able to address all of these tasks simultaneously: $\mathbf{M}(x) \to \widehat{\boldsymbol{y}} = (\widehat{y}^1, \widehat{y}^2, \cdots, \widehat{y}^n)$, where $x$ is an input image and $\widehat{y}^i$ is the prediction for task $T_i$. Here we focus on the type of per-pixel level prediction tasks (*e.g.*, semantic segmentation or depth estimation). We treat image classification as a special task, which provides global semantic description (*i.e.*, scene labels) of images and only requires image-level category annotation $c$. Therefore, the set of fully annotated real data is denoted as $\mathcal{S}_{\text{real}} = \{(x_j, y_j^1, y_j^2, \cdots, y_j^n, c_j)\}$.

**Generative learning:** Meanwhile, we aim to learn a generative model $\mathbf{G}$ that produces a set of synthesized data but with only corresponding image-level scene labels (weak annotation): $\mathbf{G}(c, z) \to \widetilde{x}$, where $z$ is a random input, and $\widetilde{x}$ is a synthesized image. The scene label of $\widetilde{x}$ is denoted as $\widetilde{c} = c$. We denote the set of synthesized images and their corresponding scene labels as $\widetilde{\mathcal{S}}_{\text{syn}} = \{(\widetilde{x}_k, \widetilde{c}_k)\}$.

**Cooperation between discriminative and generative learning:** Our objective is that the discriminative model $\mathbf{M}$ and the generative model $\mathbf{G}$ cooperate with each other to improve the performance

on the multiple visual tasks $\mathcal{T}$. During the whole process, the full model only gets access to the real fully-labeled data $\mathcal{S}_{\text{real}}$, then the generative network is trained to produce the synthesized set $\widetilde{\mathcal{S}}_{\text{syn}}$. Finally, $\mathbf{M}$ effectively learns from both $\mathcal{S}_{\text{real}}$ and $\widetilde{\mathcal{S}}_{\text{syn}}$. Note that, unlike most of the existing work on image generation (Brock et al., 2018; Zhang et al., 2019), *we do not focus on the visual realism of the synthesized images $\widetilde{x}$*. Instead, we hope $\mathbf{G}$ to capture the underlying mechanism that benefits $\mathbf{M}$.

## 3.2 FRAMEWORK AND ARCHITECTURE

Figure 2 shows the architecture of our proposed MGM framework. It contains four components: the main multi-task discriminative network $\mathbf{M}$, the image generation network $\mathbf{G}$, the refinement network $\mathbf{R}$, and the self-supervision network. By introducing the refinement network and the self-supervision network, the full model can leverage both fully-labeled real images and weakly-labeled synthesized images to facilitate the learning of latent feature representation. These two networks thus allow $\mathbf{M}$ and $\mathbf{R}$ to better cooperate with each other. Notice that our MGM is a *model-agnostic* framework, and here we instantiate its components with state-of-the-art models. In the ablation study (Sec. 4.3) and the supplementary material, we show that our MGM works well with different choices of the model components.

**Multi-task Network (M):** The multi-task network aims to make predictions for multiple target tasks based on an input image. Consistent with the most recent work on multi-task learning, we instantiate an encoder-decoder based architecture (Zamir et al., 2018; Zhang et al., 2019; Sun et al., 2019). Considering the trade-off between model complexity and performance, we use a shared encoder $\mathbf{E}$ to extract features from input images, and individual decoders for each target task. We adopt a ResNet-18 (He et al., 2016) for the encoder and symmetric transposed decoders following Zamir et al. (2018). For each task, we have its own loss function to update the corresponding decoder and the shared encoder.

**Image Generation Network (G):** The generative model $\mathbf{G}$ is a variant of generative adversarial networks (GANs). We include the generator in our framework, but this module also has a discriminator during its own training. $\mathbf{G}$ takes as input a latent vector $z$ and a category label $c$, and synthesizes an image belonging to category $c$. Considering the trade-off between performance and training cost, we instantiate $\mathbf{G}$ with self-attention generative adversarial network (SAGAN) (Zhang et al., 2019). We achieve conditional image generation by applying conditional batch normalization (CBN) layers (De Vries et al., 2017):

$$\text{CBN}\left(f_{i,c,h,w} \mid \gamma_c, \beta_c\right) = \gamma_c \frac{f_{i,c,w,h} - \text{E}\left[f_{\cdot,c,\cdot,\cdot}\right]}{\sqrt{\text{Var}\left[f_{\cdot,c,\cdot,\cdot}\right] + \epsilon}} + \beta_c, \tag{1}$$

where $f_{i,c,h,w}$ is an extracted $c$-channel 2D feature for the $i$-th sample, and $\epsilon$ is a small value to avoid collapse. $\gamma_c$ and $\beta_c$ are two parameters to control the mean and variance of the normalization, which are learned by the model for each class. We use hinge loss for the adversarial training. Notice that the proposed framework is flexible with different generative models, and we also show the effectiveness of using DCGAN (Radford et al., 2015) in the supplementary.

**Refinement Network (R):** As one of our key contributions, we introduce the refinement network $\mathbf{R}$ to further refine the shared representation using the global scene category labels. $\mathbf{R}$ takes the predictions of the multi-task network as input and predicts the category label of the input image. Importantly, because $\mathbf{R}$ only requires category labels, it can be effortlessly operationalized on the "weakly-annotated" synthesized images. Meanwhile, $\mathbf{R}$ also enforces the semantic consistency of the synthesized images with $\mathbf{G}$.

We apply an algorithm inspired by Expectation-Maximum (EM) (Dempster et al., 1977) to train the refinement network $\mathbf{R}$. For the fully-annotated real images $(x, \boldsymbol{y}, c)$, we use the scene classification loss to update $\mathbf{R}$ and refine the encoder $\mathbf{E}$ in the multi-task network $\mathbf{M}$. Then for the synthesized images $(\widetilde{x}, \widetilde{c})$, since their multi-task predictions produced by $\mathbf{M}$ might not be reliable, we only refine $\mathbf{E}$ with $\mathbf{R}$ frozen using the scene classification loss. Through refining the share feature representation with the synthesized images, this process also provides implicit guidance to the image generation network.

More specifically, we model the whole multi-task network and refinement network as a joint probability graph:

$$P(x, \boldsymbol{y}, c; \theta, \theta') = P(x) \left(\prod_{i=1}^{n} P\left(y^i \mid x; \theta\right)\right) P(c \mid \boldsymbol{y}; \theta'), \tag{2}$$

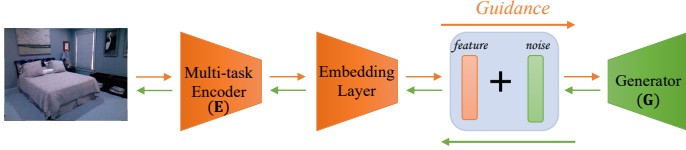

Figure 3: Joint training of the multi-task network and the image generation network. The multi-task network provides useful feature representation to guide the image generation process, while the generation network refines the shared representation through back-propagation.

---

**Algorithm 1:** The training procedure of MGM

---

**Initialization:**
$max\_epoch$: Maximum epoch for the training;
$M$: Multi-task Network, $G$: Image Generation Network;
$E$: Multi-task Encoder, $R$: the Refinement Network;
$E_{\text{self}}$: Self-supervision Network Encoder;
$N$: minibatch size;
**for** $epoch \leftarrow 1$ **to** $max\_epoch$ **do**
 Split $\mathcal{S}_{\text{real}}$ into minibatches with size $N$: $\mathcal{S}_{\text{mini}}$;
 **for** $(x, \boldsymbol{y}, c) \in \mathcal{S}_{\text{mini}}$ **do**
  $\hat{\boldsymbol{y}} = \mathbf{M}(x)$
  $\mathcal{L}_{multi}(\boldsymbol{y}, \hat{\boldsymbol{y}}) \rightarrow$ update $\mathbf{M}$;
  $\hat{c} = \mathbf{R}(\hat{\boldsymbol{y}})$
  $\mathcal{L}_{CE}(c, \hat{c}) \rightarrow$ update $\mathbf{R}$, $\mathbf{E}$;
  Sample $2N$ augmented images $x_{\text{aug}}$
  $\mathcal{L}_{NT-Xent}(x_{\text{aug}}) \rightarrow$ update $\mathbf{E}$, $\mathbf{E}_{\text{self}}$;
  Use $\mathcal{L}_{GAN}$ to train $\mathbf{G}$;
  $(\widetilde{x}, \widetilde{c}) = \mathbf{G}(\mathbf{x}, c)$
  $\mathcal{L}_{CE}(\widetilde{c}, \mathbf{R}(\mathbf{M}(\widetilde{x}))) \rightarrow$ update $\mathbf{E}$
  Sample $2N$ augmented images for the synthesized data $\widetilde{x}_{\text{aug}}$
  $\mathcal{L}_{NT-Xent}(\widetilde{x}_{\text{aug}}) \rightarrow$ update $\mathbf{E}$.
 **end**
**end**

---

where $x$ is an input image, $\boldsymbol{y}$ is the vector of multi-task predictions, $c$ is the scene label, $\theta$ is the vector of parameters of the multi-task network, and $\theta'$ is the vector of parameters of the refinement network. The parameters $\theta$ and $\theta'$ are learned to maximize the joint probability. For data samples in $\mathcal{S}_{\text{real}}$, we maximize the joint probability and update $\theta'$ to train the refinement network.

$$\theta'^{\star} = \underset{\theta'}{\operatorname{argmax}} \, P(\widetilde{c}_k \mid \boldsymbol{y}; \theta'). \tag{3}$$

For data samples in $\widetilde{\mathcal{S}}_{\text{syn}}$, we update the parameters of $\mathbf{M}$ ($\theta$) in an EM-like manner. During the $\mathbf{E}$ step, we estimate the latent multi-task ground-truth by:

$$\boldsymbol{y}^{\dagger} = \underset{\boldsymbol{y}}{\operatorname{argmax}} \, P(\boldsymbol{y} \mid \widetilde{x}_k; \theta) \, P(\widetilde{c}_k \mid \boldsymbol{y}; \theta'). \tag{4}$$

Then for the $\mathbf{M}$ step, we back-propagate the error between $\boldsymbol{y}^{\dagger}$ and $\widehat{\boldsymbol{y}}$ (the multi-task predictions) to the multi-task encoder.

$$\theta^{\star} = \underset{\theta}{\operatorname{argmax}} \, P\left(\boldsymbol{y}^{\dagger} \mid \widetilde{x}_k; \theta\right). \tag{5}$$

We use cross-entropy as the classification loss function.

**Self-supervision Network:** The self-supervision network facilitates the representation learning of the encoder $\mathbf{E}$ by performing self-supervised learning tasks on images without any annotation so that can be operationalized on both real and synthesized images. We modify SimCLR (Chen et al., 2020), one of the state-of-the-art approaches, as our self-supervision network.

This network contains an additional embedding network $\mathbf{E}_{\text{self}}$, working on the output of the multi-task encoder $\mathbf{E}$, to obtain a 1D latent feature of the input image: $\mu = \mathbf{E}_{\text{self}}(\mathbf{E}(x))$. Then, it performs contrastive learning with these latent vectors. Specifically, given a minibatch of $N$ images, this network first randomly samples two transformed views of each source image as augmented images (See supplementary material for the detailed transformations), resulting in $2N$ augmented images. For each augmented image, there is only one pair of positive augmented examples from the same source image, and other $2(N-1)$ negative pairs. Then the network jointly minimizes the distance of positive pairs and maximizes the distance of negative pairs in the latent space, through the normalized

| | Data Setting | 100% Data Setting | | | 50% Data Setting | | | | 25% Data Setting | | | |
|---|---|---|---|---|---|---|---|---|---|---|---|---|
| | Models | ST | MT | MGM | ST | MT | MGM | $MGM_r$ | ST | MT | MGM | $MGM_r$ |
| NYU v2 | SS-mIOU(↑) | 0.249 ± 0.008 | 0.256 ± 0.005 | **0.264** ± **0.005** | 0.230 ± 0.009 | 0.237 ± 0.006 | **0.251** ± **0.005** | *0.258* ± *0.004* | 0.199 ± 0.004 | 0.207 ± 0.007 | **0.229** ± **0.004** | *0.231* ± *0.005* |
| | DE-mABSE(↓) | 0.748 ± 0.019 | 0.708 ± 0.021 | **0.698** ± **0.014** | 0.837 ± 0.017 | 0.819 ± 0.018 | **0.734** ± **0.011** | *0.723* ± *0.010* | 0.908 ± 0.017 | 0.874 ± 0.015 | **0.844** ± **0.011** | *0.821* ± *0.009* |
| | SN-mAD(↓) | 0.273 ± 0.06 | 0.283 ± 0.008 | **0.255** ± **0.010** | 0.309 ± 0.008 | 0.291 ± 0.010 | **0.273** ± **0.009** | *0.270* ± *0.006* | 0.312 ± 0.007 | 0.296 ± 0.007 | **0.277** ± **0.006** | *0.274* ± *0.005* |
| Tiny Task-onomy | SS-mLoss(↓) | 0.111 ± 0.002 | 0.137 ± 0.003 | **0.106** ± **0.003** | 0.120 ± 0.003 | 0.138 ± 0.002 | **0.114** ± **0.003** | *0.112* ± *0.002* | 0.119 ± 0.003 | 0.141 ± 0.002 | **0.117** ± **0.002** | *0.115* ± *0.002* |
| | DE-mLoss(↓) | 1.716 ± 0.006 | 1.584 ± 0.008 | **1.472** ± **0.006** | 1.768 ± 0.007 | 1.595 ± 0.009 | **1.499** ± **0.008** | *1.378* ± *0.007* | 1.795 ± 0.010 | 1.692 ± 0.008 | **1.585** ± **0.009** | *1.580* ± *0.008* |
| | SN-mLoss(↓) | 0.155 ± 0.003 | 0.153 ± 0.003 | **0.145** ± **0.002** | 0.157 ± 0.002 | 0.156 ± 0.002 | **0.147** ± **0.002** | *0.140* ± *0.001* | 0.154 ± 0.002 | 0.152 ± 0.002 | **0.148** ± **0.003** | *0.142* ± *0.002* |

Table 2: Main results (mean ± std) on the NYUv2 and Tiny-Taskonomy datasets. SS: semantic segmentation; DE: depth estimation; SN: surface normal prediction. ↑ means higher is better; ↓ means lower is better. We use different metrics on the two datasets, following existing protocol. Our MGM consistently and significantly outperforms both single-task (ST) and multi-task (MT) baselines, *even reaching the performance upper-bound of training with weakly annotated real images* (MGM$_r$).

| | Data Setting | 100% Data Setting | | | 50% Data Setting | | | 25% Data Setting | | |
|---|---|---|---|---|---|---|---|---|---|---|
| | Models | MGM$_{/G}$ | MGM$_{/j}$ | MGM | MGM$_{/G}$ | MGM$_{/j}$ | MGM | MGM$_{/G}$ | MGM$_{/j}$ | MGM |
| NYU v2 | SS-mIOU(↑) | 0.261 | 0.262 | 0.264 | 0.243 | 0.243 | 0.251 | 0.215 | 0.220 | 0.229 |
| | DE-mABSE(↓) | 0.707 | 0.701 | 0.698 | 0.799 | 0.763 | 0.734 | 0.868 | 0.860 | 0.844 |
| | SN-mAD(↓) | 0.262 | 0.259 | 0.255 | 0.287 | 0.281 | 0.273 | 0.292 | 0.286 | 0.277 |
| Tiny Task-onomy | SS-mLoss(↓) | 0.108 | 0.108 | 0.106 | 0.116 | 0.115 | 0.114 | 0.119 | 0.121 | 0.117 |
| | DE-mLoss(↓) | 1.491 | 1.488 | 1.472 | 1.527 | 1.523 | 1.499 | 1.636 | 1.616 | 1.585 |
| | SN-mLoss(↓) | 0.151 | 0.151 | 0.145 | 0.153 | 0.152 | 0.147 | 0.154 | 0.152 | 0.148 |

Table 3: Comparison of our MGM model with its variants. MGM$_{/G}$: *without* generating synthesized images; MGM$_{/j}$: *without* joint learning. Our MGM outperforms single-task and multi-task baselines *even without synthesized data*, showing its effectiveness as a general multi-task learning framework. The model performance further improves with joint learning.

temperature-scaled cross-entropy (*NT-Xent*)) loss (Chen et al., 2020):

$$\ell_{i,j} = -\log \frac{\exp\left(\text{dis}\left(\mu_i, \mu_j\right)/\tau\right)}{\sum_{k=1}^{2N} \mathbb{1}_{[k \neq i]} \exp\left(\text{dis}\left(\mu_i, \mu_k\right)/\tau\right)}, \quad (6)$$

where $\ell_{i,j}$ is the *NT-Xent* loss for a positive pair of examples in the latent space $(\mu_i, \mu_j)$. $\mathbb{1}_{[k \neq i]} \in 0, 1$ is an indicator function evaluating to 1 if $k \neq i$, and $\tau$ is a temperature parameter. $\text{dis}\left(\mu_i, \mu_j\right)$ is a distance function, and we use cosine distance following Chen et al. (2020). This loss is further back-propagated to refine the multi-task encoder **E**. Notice that other types of self-supervised tasks are applicable as well. To demonstrate this, in Sec. 4.3 we also report the result with another task — image reconstruction.

## 3.3 Interaction Among Networks

**Cooperation Through Joint Training:** We propose a simple but effective joint training algorithm shown in Figure 3. The image generation network **G** takes the transferred feature representation of the multi-task encoder **E**, added with some Gaussian noise, as the latent input $z$ to conduct conditional image generation. Hence, the generation network obtains *additional, explicit guidance* (*i.e.*, extra effective features) from the multi-task network to facilitate the generation of "better images"— images that may not look more realistic but are more useful for the multiple target tasks. Then, the generation error of **G** will be back-propagated to **E** to further refine the shared representation. This process can be also viewed as introducing image generation as an additional task in the multi-task learning framework.

**Training Procedure:** We describe the procedure in Algorithm 1 and further explain it in the supplementary material.

## 4 Experiments

To evaluate our proposed MGM model and investigate the impact of each component, we conduct a variety of experiments on two standard multi-task learning datasets. We also perform detailed analysis and ablation studies.

## 4.1 Datasets and Compared Methods

**Datasets:** Following the work of Sun et al. (2019) and Standley et al. (2020), we mainly focus on three representative visual tasks in the main experiments: semantic segmentation (SS), surface normal prediction (SN), and depth estimation (DE). At the end of this section, we will show that our approach is scalable to an additional number of tasks. We evaluate all the models on two widely-

Figure 4: Visualization and error comparison of the multi-task prediction outputs in the 50% data setting. The prediction results of MGM is quite close to the ground-truth, significantly outperforming the state-of-the-art results.

benchmarked datasets: **NYUv2** (Nathan Silberman & Fergus, 2012; Eigen & Fergus, 2015) and **Tiny-Taskonomy** (Zamir et al., 2018). See Sec. B in the appendix for more details.

**Compared Methods:** We mainly focus on our comparison with two state-of-the-art discriminative baselines: **Single-Task (ST)** model follows the architecture of Taskonomy single task network (Zamir et al., 2018), and address each task individually; **Multi-Task (MT)** model refers to the sub-network for the three tasks of interest in Standley et al. (2020). These two baselines can be viewed as using our multi-task network without the proposed refinement, self-supervision, and generation networks. Note that *our work is the first that introduces generative modeling for multi-task learning, and there is no existing baseline in this direction*.

Our **MGM** is the full model trained with both fully-labeled *real* data and weakly-labeled *synthesized* data, which are produced by the generation network through joint training. In addition, to further validate the effectiveness of our **MGM** model, we consider its variant model **MGM**$_r$ that is trained with both fully and weakly labeled *real* data. **MGM**$_r$ is used to show *the performance upper bound* in the semi-supervised learning scenario, where the synthesized images are replaced by the real images in the dataset. The resolution is set to 128 for all the experiments. For all the compared methods, we use a ResNet-18 like architecture to build the encoder and use the standard decoder architecture of Taskonomy (Zamir et al., 2018).

**Data Settings:** We conduct experiments with three different data settings: (1) 100% data setting; (2) 50% data setting; and (3) 25% data setting. For each setting, we use 100%, 50%, or 25% of the entire labeled training set to train the model. For **MGM**$_r$, we add another 50% or 25% of weakly-labeled real data in the last two settings. For **MGM**, we include the same number of weakly-labeled synthesized data in all three settings.

**Evaluation Metrics:** For NYUv2, following the metrics in Eigen & Fergus (2015); Sun et al. (2019), we measure the mean Intersection-Over-Union (mIOU) for the semantic segmentation task, the mean Absolute Error (mABSE) for the depth estimation task, and the mean Angular Distance (mAD) for the surface normal estimation task. For Tiny-Taskonomy, we follow the evaluation metrics of previous work (Zamir et al., 2018; Standley et al., 2020; Sun et al., 2019) and report the averaged loss values on the test set.

**Implementation Details:** See appendix Sec. A for the training details and the sensitivity of the hyper-parameters.

## 4.2 MAIN RESULTS

**Quantitative Results:** We run all the models for 5 times and report the averaged results and the standard deviation on the two datasets in Table 2. From this table, we have the following key observations that support the effectiveness of our approach. (1) Existing discriminative multi-task learning approaches may not consistently benefit all the three individual tasks. However, our MGM consistently and significantly outperforms both the single-task and multi-task baselines across all the scenarios. (2) By using weakly-labeled synthesized data, the results of our model in the 50% data setting are even better than those of baselines in the 100% data setting. (3) More interesting, the performance of our MGM is close to MGM$_r$, which indicates that our synthesized images are *comparably useful* as real images for improving multiple visual perception tasks. (4) The performance gap between the two models is especially minimal in the 25% labeled data setting, suggesting that our proposed MGM model is, in particular, helpful with limited data.

**Qualitative Results:** We also visualize the prediction results on the three tasks for ST, MT, and MGM in the 50% data setting in Figure 4. While obvious defects can be found for all the baselines, the results of our proposed method are quite close to the ground-truth.

**How Does Generative Modeling Benefit Multi-tasks?** To have a better understanding of how the generative modeling and joint learning mechanism benefit multi-task visual learning, we also

| Model | SS-mIOU ($\uparrow$) | DE-mABSE ($\downarrow$) | SN-mAD ($\downarrow$) |
|---|---|---|---|
| MGM$_{/\text{self}}$ | 0.239 | 0.776 | 0.279 |
| MGM$_{/\text{refine}}$ | **0.254** | 0.808 | 0.290 |
| MGM$_{\text{recon}}$ | 0.241 | 0.768 | 0.285 |
| MGM | 0.251 | **0.734** | **0.273** |

Table 4: Ablation study. (1) MGM$_{/\text{self}}$: *without* self-supervision task; (2) MGM$_{/\text{refine}}$: *without* classification refinement network; and (3) MGM$_{\text{recon}}$: *with* a simple reconstruction task as self-supervision. The two proposed components are complementary and both benefit the multiple tasks. The refinement network works better for surface normal; the self-supervision network works better for semantic segmentation. Their combination achieves the best.

consider two variants of our MGM model and evaluate their performance. **MGM$_{/\text{G}}$** is the MGM model trained with $\mathcal{S}_{\text{real}}$ only (without generative modeling), which shows the performance of our proposed multi-task learning framework in general (with the help from the auxiliary refinement and self-supervision networks), and helps to understand the gain of leveraging generative modeling. **MGM$_{/\text{j}}$** is trained with synthesized images produced by a pre-trained SAGAN *without* the joint training mechanism. Table 3 shows the results on the two datasets.

Combining the results of Tables 3 and 2, we find: (1) MGM outperforms both ST and MT baseline even without generative modeling, indicating the benefit of the self-supervised task and the refinement network. (2) By introducing synthesized images that are trained separately, the multi-task performance slightly improves, which shows the effectiveness of involving generative modeling into multi-tasks, under the assistance of our refinement and self-supervision networks. (3) The joint learning mechanism further improves the cooperation between generative modeling and discriminative learning, thus enabling the generative model to better facilitate multi-task visual learning.

### 4.3 ABLATION STUDY

For all the experiments in this section, models are trained in the 50% data setting, unless specifically mentioned.

**Impact of Self-supervision Task and Refinement Network:** Two important components of the proposed framework are the self-supervision task and the refinement network. We evaluate their impact individually in Table 4. MGM$_{/\text{self}}$ is the model trained *without* the self-supervision task; MGM$_{/\text{refine}}$ is the model *without* the refinement network; for MGM$_{\text{recon}}$, we replace the SimCLR based self-supervision method with a weaker reconstruction task, and use Mean Square Error as the loss function.

We could see that the refinement network works better for the surface normal task, and the self-supervision task works better for the semantic segmentation task; they are complementary to each other, and combining them generally achieves the best performance. In addition, the model could still gain some benefit even when we use some weak self-supervision tasks like reconstruction, which indicates the generalizability and robustness of our MGM model.

**Number of Synthesized Images vs. Real images:** From the previous results, we have found that the synthesized images could benefit the target multi-tasks in a way similar to weakly labeled real images. To further investigate the impact of the number of synthesized images, we vary if from 25% to 125% during multi-task training on NYUv2 in the 25% real data setting. Figure 5 summarizes the result. First, we can see that the performance gap between MGM$_{/\text{j}}$ (without joint training) and MGM becomes larger for a higher ratio of weakly labeled data, which indicates the importance of our joint learning mechanism. *More importantly*, while the real images are constrained in number due to the human collection effort, our generation network is able to synthesize *unlimited* amounts of images. This is demonstrated in the comparison between MGM$_{\text{r}}$ (with real images) and MGM: the performance of our MGM keeps improving with respect to the number of synthesized images, achieving results almost comparable to that of MGM$_{\text{r}}$ when MGM$_{\text{r}}$ uses all the available weakly labeled real images.

We also provide ablation studies on the generalizability, impact of parameters, and the resolution of images in Sec. D in the appendix.

### 4.4 EXTENSION

**Experiments with More Tasks:** MGM is also flexible and scalable with different tasks. In addition to the three tasks addressed in the main experiments, here we add three extra tasks: Edge Texture (ET), Reshading (Re), and Principal Curvature (PC), leading to six tasks in total. We evaluate the performance of all the compared models on Tiny-taskonomy in the 50% data setting, and report the mean test loss for all the tasks. The result is reported in Table 5. Again, our proposed method still outperforms state-of-the-art baselines.

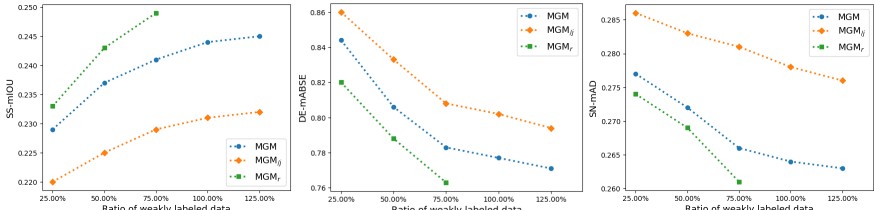

Figure 5: Performance change with different ratios of weakly labeled data. Joint learning significantly improves the performance. The performance of MGM keeps increasing with the number of the weakly labeled *synthesized* images, achieving results almost comparable to that of MGM$_r$ trained with all the available weakly labeled *real* images.

| Model | SS ($\downarrow$) | DE ($\downarrow$) | SN ($\downarrow$) | ET ($\downarrow$) | Re ($\downarrow$) | PC ($\downarrow$) |
|---|---|---|---|---|---|---|
| ST | 0.120 | 1.768 | 0.157 | 0.228 | 0.703 | 0.462 |
| MT | 0.112 | 1.747 | 0.169 | 0.241 | 0.704 | 0.436 |
| MGM | **0.108** | **1.715** | **0.152** | **0.201** | **0.699** | **0.417** |

Table 5: Mean test losses for six tasks on Tiny-Taskonomy. Again, our MGM outperforms the baselines, indicating its flexibility, generability, and scalability.

## 5 RELATED WORK

**Multi-task Learning and Task Relationship:** Multi-task learning (MTL) aims to leverage information coming from signals of related tasks so that each individual task can gain benefit (Doersch & Zisserman, 2017). Ruder (2017) identifies that most recent works use two clusters of strategies for MTL: hard parameter sharing techniques (Kokkinos, 2017; Doersch & Zisserman, 2017; Pentina & Lampert, 2017) and soft parameter sharing techniques (Misra et al., 2016; Sener & Koltun, 2018; Chen et al., 2018b). These strategies have achieved good performance for MTL with similar tasks. Researchers have also carefully studied the task relationships among different tasks to make the best cooperations among them. *Taskonomy* exploits the relationships among various visual tasks to benefit the transfer or multi-task learning (Zamir et al., 2018). Standley et al. (2020) considers task cooperation and competition, and proposes a method to assign tasks to a few neural networks to balance all of them. Some other following works also explores task relationships among different types of tasks (Sun et al., 2019; Zamir et al., 2020; Armeni et al., 2019; Pal & Balasubramanian, 2019; Yeo et al., 2021; Wallace et al., 2021). These works only consider MTL with discriminative tasks. In comparison, we first introduce generative modeling to multi-task visual learning.

**Generative Modeling for Visual Learning:** Besides the initial goal of synthesizing realistic images, some recent work has explored the potential to leverage generative models to synthesize "usefull" images for other visual tasks (Shorten & Khoshgoftaar, 2019). The most straightforward way is to generate images and the corresponding annotations as data augmentation for the target visual task (Bao et al., 2021; Sandfort et al., 2019; Choi et al., 2019; Wang et al., 2018). Another strategy to leverage generative models is through well-designed error feedback or adversarial training (Luc et al., 2016; CS Kumar et al., 2018; Mustikovela et al., 2020). There have been works that apply generative models for different visual tasks including classification (Zhan et al., 2018; Frid-Adar et al., 2018; Zhu et al., 2018), semantic segmentation (Luc et al., 2016) and depth estimation (Pilzer et al., 2018; Aleotti et al., 2018). Among them, Souly et al. (2017) also propose a semi-supervised framework to leverage generative models for semantic segmentation through adversarial training. Different from these methods, MGM is applicable to *various* multiple visual tasks and different generative networks.

**Reduced-Supervision Methods:** Recent works take advantage of weakly labeled data by assigning some self-created labels (*e.g.* colorization, rotation, reconstruction) (Noroozi & Favaro, 2016; Noroozi et al., 2017; Chen et al., 2020; Dosovitskiy et al., 2014; Pathak et al., 2016). Similar self-supervised techniques have been proved useful for multi-task learning (Liu et al., 2008; Ren & Jae Lee, 2018; Doersch & Zisserman, 2017; Lee et al., 2019). Among these techniques, a famous one is the *Expectation-Maximization (EM)* algorithm (Dempster et al., 1977; Papandreou et al., 2015), which leverages the information of weakly or unlabelled data by iteratively estimating and refining their labels. We adopt a similar spirit and introduce the refinement network for MGM framework.

## 6 CONCLUSION

This paper proposes to introduce generative modeling for multi-task visual learning. The main challenge is that current generative models almost cannot synthesize both RGB images and pixel-level annotations in multi-task scenarios. We address this problem by proposing multi-task oriented generative modeling (MGM) framework equipped with the self-supervision network and the refinement network, which enable us to take advantage of synthesized images paired with image-level scene labels to facilitate multiple visual tasks. Experimental results indicate our MGM model consistently outperforms state-of-the-art multi-task approaches.

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

We summarize the content of the appendix as follows. Section A includes additional details of model architecture of our proposed multi-task oriented generative modeling (MGM) framework. Section B describes implementation details of MGM and also the dataset settings. Section C compares MGM with other state-of-the-art multi-task models and further investigates in the few-shot regime. Section D provides additional ablation study on the training strategies for the refinement network, training with higher resolution images, generalizibility of the shared feature representation, and different types of image generation networks such as DCGAN. Section E shows more prediction visualizations. Finally, Section F discusses the potential of applying MGM to multi-task learning in other specific domains.

## A  ADDITIONAL DETAILS OF MODEL ARCHITECTURE

**Multi-task Network:**  The multi-task network contains a shared encoder network and separate decoder networks for target tasks. We use a ResNet-18 (He et al., 2016) as the encoder network, and its architecture follows the standard Pytorch implementation[1]. We only change the size of the features of each layer group from $[64, 128, 256, 512]$ to $[48, 96, 192, 360]$, so as to better address our GPU memory constraints. The size of the final feature representation is $(360, 8, 8)$. For the decoder network, we use the same architecture as in Taskonomy (Zamir et al., 2018).

**Self-supervision Network:**  We adopt SimCLR (Chen et al., 2020) as our self-supervision network. SimCLR is one of the state-of-the-art self-supervised learning approaches based on instance-level discrimination tasks. Following (Chen et al., 2020), we randomly apply 5 types of transformations on a source image to obtain an augmented image. These transformations are as follows: (1) random resizing and cropping followed by resizing back to the original size; (2) random horizontal flipping with probability of 0.5; (3) random color jittering with probability of 0.5; (4) random transformation of RGB images to gray-scale images with probability of 0.2; (5) random Gaussian blur with probability of 0.5. The shape of the transformed latent feature, which is used to perform contrastive learning, is $(128, )$.

**Refinement Network:**  The refinement network takes as input the prediction results of the multi-task network. For each individual prediction, we apply a ResNet-10 (He et al., 2016) as the refinement encoder to extract the features. The feature dimension of each layer group of the refinement encoder is the same as the multi-task encoder network. Then we concatenate all the features together and apply a fully-connected layer with the hidden size of 128 to obtain the final scene class prediction.

**Image Generation Network:**  We have instantiated the image generation network with two widely used generative networks: self-attention GAN (SAGAN)[2] (Zhang et al., 2019) in the main paper and deep convolutional GAN (DCGAN)[3] (Radford et al., 2015) in Section D.5 of this document. For DCGAN, we change the original batch-normalization layers to conditional batch normalization layers (De Vries et al., 2017) to allow conditional image generation. For the additional embedding layer used for joint learning, we use a simple global averaged pooling layer followed by a dense layer. *e.g.*

## B  IMPLEMENTATION DETAILS

**Data Processing:**  For the NYUv2 (Nathan Silberman & Fergus, 2012) dataset, following Sun et al. (2019) we resize and normalize the RGB images to $(-1, 1)$, standardize the normal ground-truth to $(0, 1)$, and do not normalize the depth. For the Taskonomy dataset (Zamir et al., 2018), we follow the standard data normalization in Zamir et al. (2018).

**Additional Implementation Details:**  We use Adam (Kingma & Ba, 2014) optimizer for all the models. The learning rates are set to 0.001 for the multi-task, self-supervision, and refinement networks, 0.0001 for the SAGAN generator, and 0.0004 for the SAGAN discriminator. The batch size is set to 32. We use a cross-entropy loss for semantic segmentation and the scene classification task of the refinement network, and an $l_1$ loss for surface normal and depth estimation.

---

[1]https://github.com/pytorch/vision/blob/master/torchvision/models/resnet.py

[2]https://github.com/voletiv/self-attention-GAN-pytorch

[3]https://github.com/Natsu6767/DCGAN-PyTorch

Due to the different converge time for the different modules, we use a three-stage strategy to perform joint training: (1) We first train the multi-task network *separately* with fully labeled real data; (2) We then freeze the multi-task network, and train the image generation network and the embedding network *separately*; (3) We do joint training with *the whole* network using both fully labeled real data and weakly labeled sythesized data. During the pre-training process of SAGAN, we set the batch size to 128 to train a better model following Zhang et al. (2019). Then, for the joint training, we use a batch size of 32 for all the sub-networks. Additionally, for the same minibatch of data, we update the image generation network 2 times iteratively during stage (3).

**Training Procedure:** We summarized the training procedure in Algorithm 1 in the main paper. Here we further explain the training procedure in more details. Given a minibatch of data in $\mathcal{S}_{\text{real}}$, we conduct the following training procedure.

1. For the input images $x$, we predict $\hat{y} = \mathbf{M}(x)$, and then use the task-specific losses between $y$ and $\hat{y}$ to update the multi-task network $\mathbf{M}$.

2. We predict the scene labels by $\hat{c} = \mathbf{R}(\hat{y})$, and update the refinement network $\mathbf{R}$ and the multi-task encoding network $\mathbf{E}$ using the cross-entropy loss between $c$ and $\hat{c}$.

3. We randomly sample pairs of augmented images, process them with the self-supervision network, and then update the self-supervision network and the multi-task encoder $\mathbf{E}$ with the *NT-Xent* loss in Eqn. (6).

4. We train the image generation network $\mathbf{G}$ through adversarial training with $(x, c)$, and back-propagate the adversarial error and update $\mathbf{E}$ at the same time.

5. We sample another minibatch of synthesized data $(\widetilde{x}, \widetilde{c})$, and use these data to update $\mathbf{E}$ by performing both the EM-like algorithm described in Section 3.2 (main paper) with $\mathbf{R}$ and the self-supervised learning as in step 3.

### B.1 Dataset Setting

We evaluate all the models on two widely-benchmarked datasets: **NYUv2** (Nathan Silberman & Fergus, 2012; Eigen & Fergus, 2015) containing 1,449 images with 40 types of objects (Gupta et al., 2013); **Tiny-Taskonomy** which is the standard tiny split of the Taskonomy dataset (Zamir et al., 2018).

For Tiny-Taskonomy dataset, since a certain amount of images for each category is required to train a generative network, we keep the images of the top 35 scene categories on Tiny-Taskonomy, with each one consisting of more than 1,000 images. This resulting dataset contains 358,426 images in total. For NYUv2, we randomly select 1,049 images as the full training set and 200 images each as the validation/test set. For Tiny-Taskonomy, we randomly pick 80% of the whole set as the full training set and 10% each as the validation/test set.

## C Additional Comparisons

### C.1 Comparison with Other Multi-task Models

In the main paper, for a fair comparison we focused on comparing our MGM model with internal models (*e.g.*, the multi-task model upon which MGM builds). To have a more comprehensive understanding of the performance of MGM, we also compare our method with some state-of-the-art multi-task models. We focus on the 25% data setting for the NYUv2 dataset, where collaboration between different tasks and the utilization of data is vitally important.

We include six models in this experiments. **ST** is the single-task model, where the encoder and the decoder are adopted from Zamir et al. (2018). **MT** is the multi-task model that uses a shared encoder as ST and separate decoders. **ST** and **MT** are the baselines compared in the main paper. **TaskGrouping** uses the optimal network for the three tasks concluded from Standley et al. (2020). Another two well-performing multi-task models are **Cross-stitch** (Ren & Jae Lee, 2018)[4] and **AdaShare** (Sun et al., 2019). **MGM** is our proposed model. Table 6 summarizes the results. *Notably, with a simple shared*

---

[4]We modify the network architecture following Sun et al. (2019) to make it work for the three tasks.

| Method | SS-mIOU($\uparrow$) | DE-mABSE($\downarrow$) | SN-mAD($\downarrow$) |
|---|---|---|---|
| ST (Zamir et al., 2018) | 0.199 | 0.908 | 0.312 |
| MT (Zamir et al., 2018) | 0.207 | 0.874 | 0.296 |
| TaskGrouping (Standley et al., 2020) | 0.215 | 0.853 | 0.292 |
| Cross Stitch (Ren & Jae Lee, 2018) | 0.205 | 0.917 | 0.296 |
| AdaShare (Sun et al., 2019) | 0.211 | 0.875 | 0.289 |
| MGM | **0.229** | **0.844** | **0.277** |

Table 6: Comparison with state-of-the-art multi-task models in the 25% data setting on the NYUv2 dataset. Notably, with a simple shared encoder architecture, our MGM model outperforms other state-of-the-art multi-task networks with more sophisticated architectures, which indicates the benefit of introducing generative modeling for multi-task learning. In addition, our MGM is a model-agnostic framework and could be incorporated with these different multi-task models for further improvement.

| Method | SS-mIOU($\uparrow$) | DE-mABSE($\downarrow$) | SN-mAD($\downarrow$) |
|---|---|---|---|
| ST | 0.162 | 1.004 | 0.337 |
| MT | 0.185 | 0.930 | 0.311 |
| MGM | **0.197** | **0.911** | **0.291** |

Table 7: Comparison in the few-shot regime – in the 10% data setting on the NYUv2 dataset where around 3 images for each scene is used as the training set. Again, MGM significantly outperforms the compared models, showing the benefit of generative models in the *extremely low-data regime*.

*encoder architecture, our MGM model outperforms other state-of-the-art multi-task networks with more sophisticated architectures, which indicates the benefit of introducing generative modeling for multi-task learning.* In addition, our MGM is a *model-agnostic framework* and could be incorporated with these different multi-task models for further improvement.

## C.2 EXPERIMENTS WITH FEW-SHOT SETTING

Since the learned generative model facilitates flow of knowledge across tasks and provides meaningful variations in existing images, it is especially beneficial in the low-data regime. So we designed a 10% data setting for NYUv2 dataset, where around 3 images for each scene is used as the training set. We also compare our MGM model with ST and MT. From Table 7, we can see that MGM outperforms the other compared models significantly, indicating the gain of generative models in the *extremely low-data regime*.

## D ADDITIONAL ABLATION STUDY

### D.1 IMPACT OF PARAMETERS

Introducing the refinement, self-supervision, and image generation networks also leads to more parameters. To validate that the performance improvements come from the novel design of our architecture rather than merely increasing the number of parameters, we provide two model variants as additional baselines: $ST_1$ and $MT_1$ use ResNet-34 as the encoder network and the corresponding decoder networks. These two networks have a similar amount of parameters as MGM. The result in Table 8 show that simply increasing the number of parameters cannot significantly boost performance.

### D.2 TRAINING STRATEGIES FOR REFINEMENT NETWORK

In the main paper, we proposed an Expectation-Maximum (EM) like algorithm to coordinate the training between the refinement network and the main network. Here we compare our **EM-Like Training (EML)** with two alternative ways of the training procedure: **Plain End-to-End Training (PEoE)** backwards the refinement loss to update the entire network directly; **Loosely Separate Training (LSeT)** trains the refinement network and the main network separately, and *only* backwards the error to the encoder network when dealing with weakly labeled images. Table 9 shows the comparison results on the NYUv2 dataset in the 50% data setting. From this table, we could find: (1) A *naïve* end-to-end training strategy is not able to facilitate the cooperation between different networks and thus hurts the overall performance; (2) Loosely separate training enables the communication

| Model | SS-mIOU ($\uparrow$) | DE-mABSE ($\downarrow$) | SN-mAD ($\downarrow$) |
|-------|----------|-----------|----------|
| ST | 0.230 | 0.837 | 0.309 |
| MT | 0.237 | 0.819 | 0.291 |
| $ST_1$ | 0.232 | 0.841 | 0.304 |
| $MT_1$ | 0.236 | 0.804 | 0.288 |

Table 8: Impact of parameters. $ST_1$ and $MT_1$: baselines with a larger number of parameters (with deeper backbones). Simply increasing the number of parameters cannot significantly boost performance.

| Method | SS-mIOU($\uparrow$) | DE-mABSE($\downarrow$) | SN-mAD($\downarrow$) |
|--------|----------|-----------|----------|
| MT | 0.237 | 0.815 | 0.291 |
| PEoE | 0.211 | 0.896 | 0.301 |
| LSeT | 0.247 | 0.768 | 0.277 |
| EML | **0.251** | **0.734** | **0.273** |

Table 9: Results on the NYUv2 dataset in the 50% data setting *with different training strategies for the refinement network*. 'MT': multi-task learning baseline; 'PEoE': plain end-to-end training; 'LSeT': loosely separate training; 'EML': EM-like training (proposed in the main paper). Our EML significantly outperforms alternative strategies to train the refinement network.

between the refinement network and the main network *when needed* and thus outperforms the baseline; (3) Our proposed EM-like training strategy further improves over the loosely separate training and achieves the best performance.

### D.3 EXPERIMENTS ON HIGHER IMAGE RESOLUTION

In principle, the proposed framework is agnostic to the specific types of multi-task networks and image generation networks, thus flexible with image resolutions. The practical constraint lies in that it is still challenging and resource-consuming for modern generative models to synthesize very high-resolution images (Zhang et al., 2019; Brock et al., 2018), although the deep multi-task models normally work better with high-resolution images. In the main experiments, consistent with exiting image synthesis work (Zhang et al., 2019), we focused on the resolution of $128 \times 128$. Here we further made an attempt to run our experiments with a higher resolution, $256 \times 256$ on NYUv2. The results of all the compared models in the 50% data setting are shown in Table 10. We could find that MGM still consistently outperforms the baselines, indicating the great robustness and flexibility of our proposed framework.

### D.4 GENERALIZATION OF THE SHARED FEATURE REPRESENTATION

Intuitively, our MGM achieves state-of-the-art performance by effectively learning a shared feature representation. We further show the generalization capability of this representation by designing the following experiment: for the multi-task model and our MGM model, we first learn the shared feature space with the SS and DE tasks, and we then use that learned feature space to train a new decoder for the SN task. We report the results on NYUv2 in Table 11. Our MGM outperforms the multi-task model in all the three data settings, which means that MGM indeed learns a better and robust shared feature space.

### D.5 EXPERIMENTAL RESULTS WITH DCGAN

The proposed MGM model is a general framework and is flexible with different choices of model components. Here we show its flexibility for the image generation network: We replace SAGAN with DCGAN (Radford et al., 2015), and Table 12 shows the result on the NYUv2 dataset in the 50% data setting. Similar to the results in the main paper, MGM equipped with DCGAN ('MGM-DCGAN') consistently improves the performance on all the tasks, which indicates the robustness of the proposed MGM framework. In addition, we find that 'MGM-SAGAN' outperforms 'MGM-DCGAN', suggesting that a more powerful image generation network leads to better performance.

| Model | SS-mIOU ($\uparrow$) | DE-mABSE ($\downarrow$) | SN-mAD ($\downarrow$) |
|-------|-----------|------------|-----------|
| ST | 0.239 | 0.849 | 0.282 |
| MT | 0.244 | 0.834 | 0.313 |
| MGM | **0.257** | **0.819** | **0.275** |

Table 10: Experiments with 256 image resolution on the NYUv2 dataset. Our MGM still consistently outperforms the compared baselines, showing the great robustness and flexibility of the proposed framework.

| Model | mAD-100%($\downarrow$) | mAD-50%($\downarrow$) | mAD-25%($\downarrow$) |
|-------|------------|-----------|-----------|
| MT | 0.291 | 0.310 | 0.323 |
| MGM | **0.280** | **0.298** | **0.305** |

Table 11: Results for the SN task with pre-trained feature representations by the SS and DE tasks. MGM consistently outperforms multi-task (MT), indicating that MGM learns a more effective and generalizable feature representation.

## E    MORE VISUALIZATIONS

In Figure 4 of the main paper, we visualized the prediction results. Here we provide more visualizations of the multi-task predictions for MGM and the compared baselines in Figure 6. Our MGM model significantly outperforms both ST and MT baselines.

## F    DISCUSSION AND FUTURE WORK

In the main paper, we focused on applying generative networks to *general* multi-task learning. On the other hand, generative models can also be exploited to address multiple tasks in *specific* domains, such as human face modeling (Li et al., 2019; 2016; Han et al., 2017; Chen et al., 2018a), autonomous driving (Liu et al., 2020; Wang et al., 2019; Chennupati et al., 2019), and bio-medical processing (Zhou et al., 2020; Panagopoulos, 2017). We leave such exploration of leveraging our MGM model to these domains as future work.

## G    ADDITIONAL EXPERIMENTAL EVALUATIONS

### G.1    EXPERIMENTAL EVALUATION ON CITYSCAPE SUBSET

In this section, we demonstrate that MGM can work with datasets when no image-level labels are available. In an alternative may, the proposed MGM frame model can work with object labels as well since the generative network and refinement network can naturally work with multi-hot labels — the refinement network can work with a multi-label classifier, and the generative network can be a multi-label-conditional GAN.

We conducted semantic segmentation on a subset of CityScape (Cordts et al., 2016) dataset, the Zurich street scene. We focused on the semantic segmentation task, which is a representative task on CityScape. We use the 30 standard multi-hot CityScape semantic object labels for the generative model and also the refinement networks. We use 80% of the data for training and 20 % for testing and compare the performance of **ST** and **MGM** for this experiment. We generate the same amount of data with random multi-hot labels using MGM. The results are shown in Table 13. MGM still outperforms the baseline ST model, indicating the robustness and generalizability of the model with multi-hot object labels.

### G.2    EXPERIMENTS WITH EXTREME LOW DATA AT TASKONOMY

We noticed that for Tiny-Taskonomy dataset, 25% data setting is still far from low-data regime. To further explore the effectiveness with our model with low-data, we further conduct an experiment with a subset of Tiny-Taskonomy dataset. We randomly select 3 nodes (allensville, benevolence, and coffeen) from Tiny-Taskonomy dataset to build a dataset with 17,404 images—around 5% data setting compared with the full Tiny-Taskonomy dataset. We then conduct experiments with ST,

| Method | SS-mIOU(↑) | DE-mABSE(↓) | SN-mAD(↓) |
|---|---|---|---|
| ST | 0.233 | 0.835 | 0.309 |
| MT | 0.237 | 0.815 | 0.291 |
| MGM-SAGAN | **0.251** | **0.734** | **0.273** |
| MGM-DCGAN | **0.245** | **0.750** | **0.285** |

Table 12: Results on the NYUv2 dataset in the 50% data setting *with different image generation networks*. 'MGM-SAGAN': MGM equipped with SAGAN (presented in the main paper); 'MGM-DCGAN': MGM equipped with DCGAN. Both 'MGM-SAGAN' and 'MGM-DCGAN' consistently improve the performance on all the tasks and outperform single-task (ST) and multi-task (MT) baselines. This shows the robustness and flexibility of the proposed MGM framework. In addition, 'MGM-SAGAN' outperforms 'MGM-DCGAN', suggesting that a more powerful image generation network leads to better performance.

| Method | SS-mIOU(↑) |
|---|---|
| ST | 0.57 |
| MGM | **0.64** |

Table 13: Results on the CityScape Subset. MGM still outperforms the baseline ST model, indicating the robustness and generalizability of the model with multi-hot object labels.

MT and MGM for this subset. All the other experimental settings keep the same as the main paper. Table 14 shows the comparable results. Combining the results in Table 2, we can find that MGM consistently outperforms ST and MT and is robust and especially helpful in low-data regime.

### G.3 ABLATION WITH SINGLE TASKS

MGM is a general framework that can be applied to both single tasks and multiple tasks. In the main submission, we mainly focused on the more challenging multi-task scenario. In this subsection, we conduct experiments with single tasks. Here we add three baselines applying MGM to the single tasks (SS, DE, SN) named **MGM-SS**, **MGM-DE** and **MGM-SN** in the NYUv2 50% data setting. We further provide an additional baseline by using the equivalently sampled data from the above three model as the augmented data, but not jointly train the generative network, named **MGM-Combine**. The results of these models are shown in Table 15. We have the following observations: (1) The proposed MGM framework can consistently benefit each single tasks though without leveraging shared features from multiple tasks. (2) Compared with the full MGM model, the performance drops when only using single tasks to jointly train with the generative model. (3) When using the individually optimized generative models, the performance is slight better than MGM$_{/j}$ but still could not reach MGM, indicating the importance of our joint training mechanism.

## H VISUALIZATIONS OF GENERATED IMAGES

We visualize the generated images for the Tiny-Taskonomy dataset of SAGAN (Zhang et al., 2019) and MGM in Figure 7. From Figure 7, we observe that (1) the conventional SAGAN, trained with the realistic objective and without the guidance of the downstream tasks, produces photo-realistic images; (2) the visual quality of the generated images by MGM becomes degraded, where SAGAN is jointly trained with the multi-task learning objective and under the guidance of the downstream tasks. Interestingly, a similar phenomenon has been observed in (Souly et al., 2017), where a generative model is used to facilitate the semantic segmentation task. We hypothesize this is because those synthesized images that are useful for improving downstream tasks might not be necessarily photo-realistic. While the images synthesized by MGM are not visually realistic, they may contain some crucial discriminative information that can be leveraged for addressing the downstream tasks — for example, the synthesized images may contain some unseen patterns from the real images, which increases the diversity of the training data. In addition, the difference of the synthesized images between SAGAN and MGM can also partially explain the result in the pilot study — the images synthesized off the shelf are quite different from the desired images for multi-task learning, and thus they are not effective in facilitating downstream tasks.

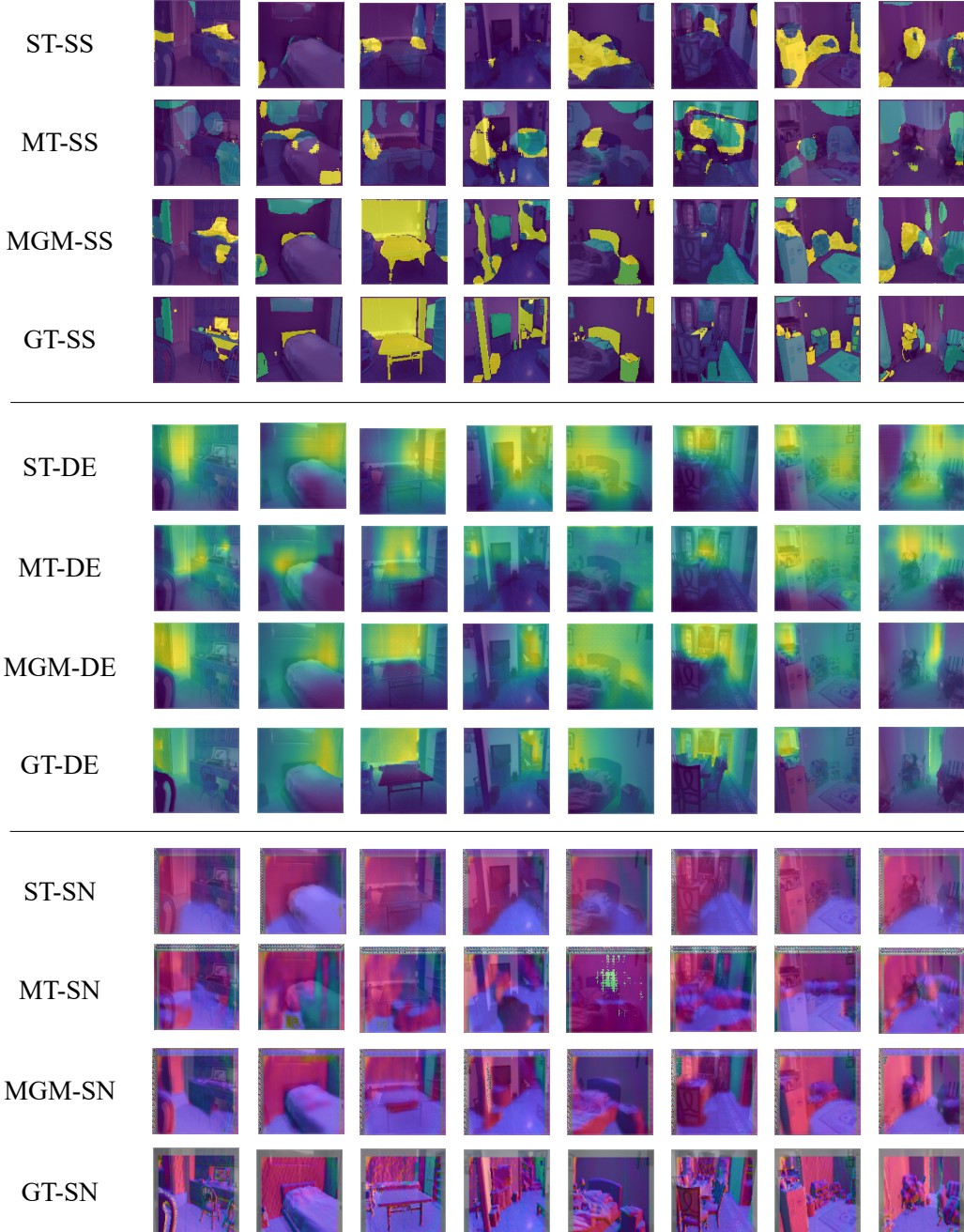

Figure 6: More visualizations of the multi-task predictions for MGM and the compared baselines. SS: semantic segmentation task; DE: depth estimation task; SN: surface normal prediction task; ST: single-task model; MT: multi-task model; MGM: multi-task oriented generative modeling (our proposed model); GT: ground-truth. The prediction results of our MGM model are much closer to the ground-truth and significantly outperform the state-of-the-art results.

## I   VISUALIZATIONS OF THE SEMANTIC ANNOTATIONS

We also visualize two samples and the corresponding semantic segmentation annotations we used for the pilot study in Figure 8. The visualization shows that the annotation is accurate and using Taskonomy is sufficient for training.

| Model | SS ($\downarrow$) | DE ($\downarrow$) | SN ($\downarrow$) |
|-------|------|------|------|
| ST | 0.137 | 1.836 | 0.161 |
| MT | 0.156 | 1.807 | 0.162 |
| MGM | **0.125** | **1.670** | **0.153** |

Table 14: Comparison with extreme low data in Taskonomy. In this data setting, MGM significantly outperforms both ST and MT, indicating that MGM is robust and especially helpful in low-data regime.

| Model | SS-mIOU ($\uparrow$) | DE-mABSE ($\downarrow$) | SN-mAD ($\downarrow$) |
|-------|------|------|------|
| ST | 0.230 | 0.837 | 0.309 |
| MGM-SS | 0.244 | - | - |
| MGM-DE | - | 0.752 | - |
| MGM-SN | - | - | 0.277 |
| MGM-Combine | 0.249 | 0.747 | 0.277 |
| MGM | **0.251** | **0.734** | **0.273** |

Table 15: Ablation with MGM for single tasks and a stronger baseline with the learned information from the three individual tasks but without jointly training. The experiments are conducted on NYUv2 50% data setting. MGM-SS, MGM-DE, MGM-SN: variantal MGM model for single tasks. MGM-Combine: MGM variant trained with augmented images generated by the above three models. The proposed MGM framework can consistently benefit each single tasks and MGM-combine cannot reach the performance of MGM, indicating the importance of joint training mechanism.

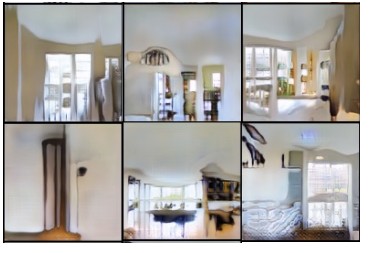 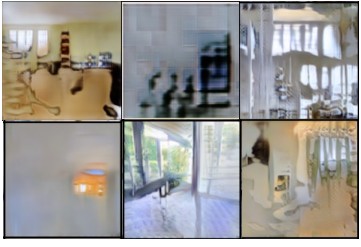

**SAGAN**  **MGM**

Figure 7: Generated images of SAGAN and MGM for the Tiny-Taskonomy dataset. After jointly training, the images are not visually realistic, but they are helpful to improve the downstream task performance.

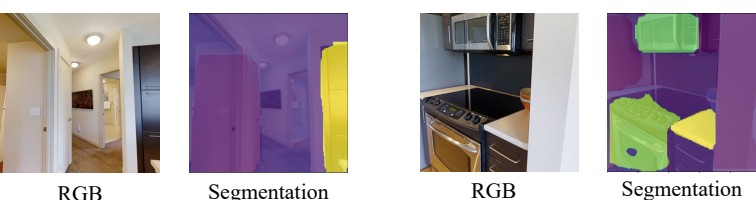

RGB    Segmentation    RGB    Segmentation

Figure 8: Paired RGB and semantic segmentation labels. This paired data indicates that the oracle annotator we used in the pilot study is effective.

