# OpenReview forum: "Generative Modeling for Multitask Visual Learning"
_ICLR.cc/2022/Conference — ICLR 2022 Submitted_

### Official Review · Reviewer_X2gh · 2021-11-02

**Correctness:** 3
**Technical Novelty And Significance:** 3
**Empirical Novelty And Significance:** 2
**Recommendation:** 5
**Confidence:** 4

**Main Review:**

Strengths:
- A pilot experiment that shows off-the-shelf image generators is not suitable as data augmentation.

- The paper is abundant with experiments and most of the numerical results shows good indication that the proposed method is better than existing state-of-the-art, and ablations studies somewhat justifies the design choices of the method.


Weakness:

Major concerns:

- Does it really help when in low data regime?

   I don't think the remark of being helpful in low data regime is convincing according to the experiments. On the NYU data set, which is ~1000 images for training, I think it is safe to say that even 100% of data is in this 'low data' regime. Therefore I don't really think the experiments on 25% or 50% of NYU is indicative of the model's performance if the original dataset is, for example, ~100k.

  For the taskonomy results, ST/MT seems to have less degradation when the training data gets more and more limited. For example, ST on semantic segmentation drops from .111 to .119, yet the proposed method drops from .106 to .117. Does this mean ST is more robust to sparser data? Also, on normal prediction, it seems that ST/MT is better at 25% of data compared to 100%. Does it mean that they are better models in the low data regime?


- Too many components, hurts the generative model narrative; The generative modeling helps, but is it absolutely necessary?

   The narrative of the paper focuses on the generative model that is jointly trained with the multitask model. But, it seems it only brings marginal improvements as shown in table 3 with MGM/G. The performance drop without the generative model is really just marginal. Yet, in table 4 and 9, it seems that the optimization scheme and the self-supervision block have more significant impact on the performance. This really makes one wonder why is the generative model necessary at all?

- Are numbers indicative?

  I would refrain from stating "our results are quite close to GT". According to the qualitative results shown in the main text and in the supplementary, it seems that the prediction quality is quite low for all methods. This somewhat hurts the interpretation of the numbers, since they are more effective at telling which results is better instead of being 'less bad'. But this could also be the visualization issue, where the RGB colors are blended in as well.

- inconsistent metrics

  I understand the authors are following the conventions for evaluation on different dataset, but it makes is hard to read the table when the metrics are different across the dataset. It would be more indicative if those are made consistent. I'm not aware of the specifics of evaluating on taskonomy, but it seems the NYU metrics are more universal.

Minor concerns:

- visualization of the results.
All the visualization of depth/normal/semantic segmentations are overlaid with the original image, which is confusing and makes it hard to compare across methods and against GT. It makes more sense to show just the original result. Also the normal seems to have weird padding around the prediction, and sometimes there's some noise pattern around the boundary. Is that a bug? if those regions are included in metric calculation, then the numbers are meaning less as those might dominate depending on the form of the metric.


- The paper also has some minor issues in writing, but it seems those are not limiting the quality of the paper at the current state.



**Summary Of The Paper:**

The paper proposes a multi-task learning framework that incorporates a generative model, which is designed as an in-the-loop data augmenter that could improve the downstream tasks, especially in the low-data regime. The authors made strong emphasis on using a generative modeling as an effective form of data augmentation during training.

The major contributions, as perceived by the reviewer, are the following:
1. Showed that using an off-the-shelf image generator does not improve the down stream tasks.
2. Proposed a jointly learning framework, which includes an in-the-loop generator, a self-supervision block, a refinement block and an EM style optimization scheme that outperforms single-task and multi-task baselines.
3. Made ablation studies that shows the importance of different design choices.


**Summary Of The Review:**

I believe this paper is slightly below the the acceptance threshold.

The paper presents a novel joint frame work that includes a generative model for multi-task learning, and quantitative experiments show that the proposed method out-performs baseline methods.
But, it is hard to interpret the results since:

1. The proposed methods include a generative modeling block, a self-supervision block and a refinement block. Through the ablation studies, it seems that the generative blocks, brings marginal benefits while other blocks such as self-supervision blocks, brings a larger margin for improvements.

2. Authors claim that the proposed method is better at low-data regime, which is a less convincing remark to give according to the experiment results: The NYU dataset is extremely small, which makes data ablation experiments on NYU really hard to interpret. Does it really make sense to train on 25% of NYU, which is ~250 images, with a deep learning network? The performance drop of ST/MT on tiny taskonomy is less significant than the proposed method, does that mean they are better at low-data regime?

3. It seems the author proposed a multitask learning framework that outperforms some baseline, but it hard to interpret why.

Based on theses observation, I believe this paper is slightly below acceptance threshold.

---

> ### Author Response · Authors · 2021-11-23
> **Response to Reviewer X2gh 1/2**
>
> We thank the reviewer for the valuable comments. The comments focus mostly on the analysis of results and visualization of results. We address all the points as follows.
>
> 1. Experiments in low-data regime:
>
> We thank the reviewer for the comment. First, we respectfully disagree with the reviewer that our model is not helpful in the low-data regime. In fact, the extensive experimental evaluation clearly shows that our MGM model is **consistently helpful across the board in different sample-size regimes**, including the challenging low-data regime.
>
> Meanwhile, we realized the confusion in the discussion on our model in the low-data regime. For clarity, we have revised the related text in the main result section and also in the introduction. Also, we agree with the reviewer that the low-data regime is a relative concept here -- the low-data regime with 50% or 25% data is compared with the original 100% data setting; the absolute number of training images for the 100% data setting varies across the benchmarks (i.e., on NYUv2 and Tiny-Taskonomy), and so for the 50% and 25% data settings. Therefore, the specific behaviors of our MGM model vary across these different data settings, but our MGM model consistently achieves the best performance.
>
> Regarding the performance gap in the 25% data setting, we think the reason is that the ST baseline optimizes a separate model for each specific task; but for MGM, it learns a single model for all the tasks, and thus it needs to deal with and balance between multiple tasks. Therefore, there might be fluctuation in performance improvement for a particular task, compared with the ST model. But again, MGM consistently outperforms ST.
>
> In addition, we would like to provide a different perspective on interpreting the performance change from the 25% to the 100% data settings. (1) In both settings, our MGM outperforms the ST baseline. (2) If we see the trend from the 25% data setting to the 100% data setting -- with the increase of the training data samples, the performance gap between MGM and ST becomes larger. This indicates that our MGM model is more powerful and scalable with respect to better use of more data.
>
> Moreover, we think different tasks may require different amounts of data to converge, due to their different levels of difficulty. For example, for the surface normal task, 25% of the full dataset may be sufficient for the ST and MT models to converge. But for MGM, since we introduce more data variances from the generative model (may also see Appendix Section H for the visualization), we can always bring additional improvements for all the tasks.
>
> Finally, as mentioned above, the low-data regime is a relative concept here -- the 25% data setting on NYUv2 and the 25% data setting on Tiny-Taskonomy correspond to different absolute numbers of examples; Tiny-Taskonomy consists of much more data. Therefore, to investigate the performance of MGM in the low-data regime of Tiny-Taskonomy (that is comparable to NYUv2), we further decrease the number of images for the Tiny-Taskonomy dataset to form a subset with 17,404 images (around 5% data setting compared with the full Tiny-Taskonomy dataset ). The results on this setting are: ST (SS 0.137; DE 1.836; SN 0.161), MT (SS 0.156; DE 1.807; SN 0.162), and MGM (SS 0.125; DE 1.670; DN 0.153), which are shown in Section G.2 and Table 14 in the Appendix. From this experiment, we can draw similar conclusions as the experiments on the NYUv2 25% data setting, for which MGM can be greatly helpful when the data is limited.

---

> > ### Author Response · Authors · 2021-11-23
> > **Response to Reviewer X2gh 2/2**
> >
> > 2. Too many components hurt the generative model narrative:
> >
> > First, we would like to argue that the main goal of this paper is to show how to train a generative model that 1) is able to take into account the downstream tasks’ supervision, and 2) facilitates the downstream tasks. We believe this provides a new perspective on multi-task learning and generative modeling. The self-supervised network and the refinement network are proposed for this purpose -- although they are effective as standalone components without the generative model, this is not our main focus here. **The self-supervised network and the refinement network make it possible to leverage generative models for multi-task visual learning with only weak annotations.**
> >
> > Second, for the performance gap between MGM/G and MGM, we think it is because MGM/G is already a strong model with the refinement network and the self-supervision network -- it is relatively hard for MGM to further improve the performance with a stronger model.
> >
> > Third, the comparison in Table 9 in the Appendix on different training schedules (PEoE vs. EML) is mainly used to evaluate the proposed EM-alike training algorithm for the refinement network. We find that an inappropriate training schedule for the refinement network will hurt the model performance, but this training schedule will not affect the model performance with generative models.
> >
> > Again, our main focus of this paper is to explore introducing generative models for multi-task visual learning, since there is no trivial method to directly apply generative models for multi-task learning. And we believe this attempt will inspire further work in the related area, such as exploiting the cooperation between generative modeling and discriminative learning. We don’t think the powerful components of the self-supervision network and the proposed refinement network will affect the main idea of introducing generative modeling for multi-task visual learning in a trivial way.
> >
> >
> > 3. Visualized multi-task predictions:
> >
> > We thank the reviewer for pointing out the visualization issue. We carefully double-checked the visualizations in the Appendix and found that there were mismatched visualizations in Figure 6, causing the confusion of the model effectivenesses. We have updated Figure 6 with the correctly matched results. We also plotted the visual predictions with a larger resolution, so that it is clearer to see the details and to capture the performance improvements of our model over both ST and MT. We apologize for the confusion caused by the previous figure.
> >
> > From the updated Figure 6, we can find that, compared with ST and MT, our MGM has the minimum error w.r.t. ground-truth. We also agree with the reviewer that the overall quality of these predictions are not perfect yet. But notice that MGM is a general framework that can be applied on top of any multi-task backbones. In our main evaluation, we instantiated MGM with a naive backbone, and MGM can get better results and improved visualizations when working on strong backbones.
> >
> > 4. Inconsistent Metrics
> >
> > We thank the reviewer for suggesting using consistent metrics, such as using the NYUv2 metrics for both the NYUv2 and Tiny-Taskonomy datasets. However, doing so will result in two issues. First, this will make it difficult to compare with prior work. As mentioned in Section 4.1, we follow the previous work on evaluating the two benchmarks. Second, it is actually difficult to extend the NYUv2 metrics to be applicable on Tiny-Taskonomy.
> >
> > This is because NYUv2 is a small dataset with only 3 tasks (SS, DE, and SN). Therefore, we adopt the commonly used metrics for the three tasks. However, the Tiny-Taskonomy dataset is a standard large-scale benchmark for multi-task visual learning, which contains more than 20 different tasks. The standard metrics on Tiny-Taskonomy are mean test loss. Evaluating with mean test loss can help us better calibrate the model performances across different tasks and analyze task relationships, regardless of the types of tasks. In addition, in Section 4.4, we investigated 6 tasks in addition to SS, DE, and SN; using mean test loss makes it convenient to analyze the results for the additional tasks. Based on these considerations, for the Tiny-Taskonomy dataset, we follow recent works on multi-task learning [Ref2, Ref3, Ref4] to adopt the mean test loss as the evaluation metric.
> >
> > [Ref2] Amir R Zamir, Alexander Sax, William Shen, Leonidas J Guibas, Jitendra Malik, and Silvio Savarese. “Taskonomy: Disentangling task transfer learning.” In CVPR, 2018.
> >
> > [Ref3] Trevor Standley, Amir Zamir, Dawn Chen, Leonidas Guibas, Jitendra Malik, and Silvio Savarese. “Which tasks should be learned together in multi-task learning?” In ICML, 2020.
> >
> > [Ref4] Ximeng Sun, Rameswar Panda, Rogerio Feris, and Kate Saenko. “Adashare: Learning what to share for efficient deep multi-task learning.” In NeurIPS, 2020.

---

> > > ### Comment · Reviewer_X2gh · 2021-11-30
> > > **Thanks for the response**
> > >
> > > I would like to thank the authors for their replies. However, I found my concerns only partially addressed.
> > > 1. My main concern is about the claim on 'low data' regime. I have no doubt that the proposed method is better than the baselines in the absolute sense, across different data percentage. It would make more sense if the authors simply claim that using a generative model is just better than baseline, regardless of data abundance. Since this 'low data' regime is a major narrative, I'm not sure if the experiments supports that claim. Though it is perfectly fine for the authors to change the narrative and suit that to the experiments, I don't think this indicates that the paper is ready enough for publication.
> > >
> > > 2.
> > > > Second, for the performance gap between MGM/G and MGM, we think it is because MGM/G is already a strong model with the refinement network and the self-supervision network -- it is relatively hard for MGM to further improve the performance with a stronger model.
> > >
> > > I find this argument way too weak to prove the design choice of this generative model. Given the qualitative results shown, I don't think the few points in the result is significant enough to prove that the generative model is essential.
> > >
> > > 3.
> > > I found fig 6 still have border artifacts, and with rgb images as background overlay. I don't think the authors addressed this issue properly
> > >
> > > 4.
> > > I don't find the necessity of keeping a consistent metrics with taskonomy given the specific task of depth/normal/semantic prediction. Taskonomy has various tasks, but since major evaluation is done on those three specific tasks, it makes more sense to report consistent metrics on theses tasks.
> > >
> > > Given those considerations, I vote for rejection for this paper.

---

> > > > ### Author Response · Authors · 2021-12-01
> > > > **Additional Response to Reviewer X2gh Part (1/2)**
> > > >
> > > > We thank Reviewer X2gh again for the time and the valuable comments. We further address the mentioned concerns as follows:
> > > >
> > > > > My main concern is about the claim on 'low data' regime. I have no doubt that the proposed method is better than the baselines in the absolute sense, across different data percentage. It would make more sense if the authors simply claim that using a generative model is just better than baseline, regardless of data abundance. Since this 'low data' regime is a major narrative, I'm not sure if the experiments supports that claim. Though it is perfectly fine for the authors to change the narrative and suit that to the experiments, I don't think this indicates that the paper is ready enough for publication.
> > > >
> > > > We thank the reviewer for the suggestion on the narrative. We respectfully disagree with the reviewer that “low data regime is a major narrative”. In fact, in the original version, “low-data regime” appeared only **3** times in the main submission; in the revised version, as the reviewer suggested, we mentioned “low-data regime” only **2** times. We did not claim that our MGM model only works in low-data regimes.
> > > >
> > > > Instead, our narrative is that the proposed MGM model is consistently helpful across the board in different sample-size regimes, even in the challenging low-data regime. We believe that our experiments are supportive of this narrative. In particular, as mentioned in the previous response, our experiments in the low-data regime (the original results in the 25% data settings on both NYUv2 and Tiny-Taskonomy, and the new results in the 5% data setting on Tiny-Taskonomy in Appendix Section G.2 and Table 4) validate that our MGM model indeed outperforms all the baselines in the challenging low-data regime. We hope the reviewer will appreciate our additional efforts under this challenging new setup (with 5% data on Tiny-Taskonomy).
> > > >
> > > >
> > > > >I find this argument way too weak to prove the design choice of this generative model. Given the qualitative results shown, I don't think the few points in the result is significant enough to prove that the generative model is essential.
> > > >
> > > > We respectfully disagree with the reviewer’s interpretation of our work. We would like to reiterate that the main goal of this work, as mentioned in the previous response, is to investigate a new problem that has not been explored before -- how to train a generative model that 1) is able to take into account the downstream tasks’ supervision, and 2) facilitates the downstream tasks. We believe this provides a new perspective on (multi-task) discriminative learning and generative learning.
> > > >
> > > > Consistent with this goal, **Our proposed self-supervised network and refinement network makes it possible to learn a generative model that benefits multi-task learning with only weak annotations.** Without these two components, the off-the-shelf generative models do not help the downstream tasks, as shown in our pilot study. That being said, although the self-supervised network and the refinement network are effective as standalone components without the generative model, this is not our main focus here -- we are not showing that the self-supervised and refinement networks are effective for multi-task learning; instead, we are showing that they enable a generative model to be helpful for multi-task learning.

---

> > > > > ### Author Response · Authors · 2021-12-01
> > > > > **Additional Response to Reviewer X2gh Part (2/2)**
> > > > >
> > > > > > I found fig 6 still have border artifacts, and with rgb images as background overlay. I don't think the authors addressed this issue properly
> > > > >
> > > > > We greatly appreciate the reviewer’s suggestion on an alternative way of visualization. We seriously considered the reviewer’s suggestion and improved our visualization, as detailed in the previous response. We indeed tried the visualization without overlaying the RGB images, but we found that led to worse visualization. The details are explained below.
> > > > >
> > > > > First, for the border noise of the surface normal, **it is not due to the visualization, but because of the annotations themselves on the dataset**. More specifically, the surface normal on NYUv2 was annotated by [Ref7]; these annotations themselves missed the border parts (See Fig. 6 `Ground-truth’, they are padded by 0). Therefore, all of the models including the ground-truth will have border noise.
> > > > >
> > > > > Second, regarding using RGB images as background overlay, (1) we have tried not overlaying the RGB images as the reviewer suggested, but the visualizations are pretty bad: we cannot find what the object is for semantic segmentation, and the depth images make no sense even for the ground-truth. (2) In fact, overlaying predictions on the top of RGB images is a popular visualization method used by previous works like [Ref5, Ref6]. Besides, the COCO dataset visualizes the segmentations on top of the RGB images.
> > > > >
> > > > > Importantly, with the current visualization, we can easily observe the significant differences between the MGM model and the ST and MT baselines. We don’t think different ways of visualization will change our conclusion on the effectiveness of our approach.
> > > > >
> > > > >
> > > > >
> > > > > > I don't find the necessity of keeping a consistent metrics with taskonomy given the specific task of depth/normal/semantic prediction. Taskonomy has various tasks, but since major evaluation is done on those three specific tasks, it makes more sense to report consistent metrics on theses tasks.
> > > > >
> > > > > We greatly appreciate the reviewer’s suggestion on using the same evaluation metric for the two datasets, i.e., using the NYUv2’s metric for Tiny-Taskonomy. However, we would like to argue that (1) using different metrics **will not** lead to different conclusions; (2) **using the NYUv2’s metric for Tiny-Taskonomy will cause other issues**, which are not easy to tackle.
> > > > >
> > > > > First, we just followed the standard metrics on Tiny-Taskonomy. Using the NYUv2’s metric for Tiny-Taskonomy will make it difficult to compare our work with prior work, and it will also make it difficult for future work on Tiny-Taskonomy to compare with our work.
> > > > >
> > > > > Second, NYUv2’s metric only applies to the three specific tasks (SS, DE, and SN); using NYUv2’s metric for Tiny-Taskonomy is not flexible and will make the comparison difficult for other types of tasks. As the reviewer also pointed out, Tiny-Taskonomy contains more tasks beyond the three tasks. In fact, as mentioned in the previous response, our MGM framework is general and can be applied to other tasks. In Table 5 in the main submission, we extended to 6 tasks on Tiny-Taskonomy. **If we were using the NYUv2’s metric, then in Table 5 we would have to apply two different types of metrics on the Tiny-Taskonomy dataset -- NYUv2’s metric for the SS, DE, and SN tasks, and Tiny-Taskonomy’s metric for the Edge Texture (ET), Reshading (Re), and Principal Curvature (PC) tasks**. We think it is better to keep the metric consistent within the same dataset.
> > > > >
> > > > > Finally, our evaluation strategy is consistent with the previous work. For example, [Ref4] also uses slightly different metrics for different datasets in their evaluation, but the conclusion drawn from different datasets is consistent.
> > > > >
> > > > >
> > > > > [Ref5] Sixiao Zheng, Jiachen Lu, Hengshuang Zhao, Xiatian Zhu, Zekun Luo, Yabiao Wang, Yanwei Fu, Jianfeng Feng, Tao Xiang, Philip H.S. Torr, and Li Zhang. "Rethinking semantic segmentation from a sequence-to-sequence perspective with transformers." In CVPR, 2021.
> > > > > [Ref6] Carion, Nicolas, Francisco Massa, Gabriel Synnaeve, Nicolas Usunier, Alexander Kirillov, and Sergey Zagoruyko. "End-to-end object detection with transformers." In ECCV, 2020.
> > > > > [Ref7] Eigen, David, and Rob Fergus. "Predicting depth, surface normals and semantic labels with a common multi-scale convolutional architecture." In CVPR, 2015.

---

### Official Review · Reviewer_Ytpa · 2021-11-03

**Correctness:** 3
**Technical Novelty And Significance:** 2
**Empirical Novelty And Significance:** 2
**Recommendation:** 5
**Confidence:** 3

**Main Review:**

The proposed framework that combines multi-task learning and generative modeling is interesting and inspiring. I also appreciate the extensive experiments (ablation studies and extensions) conducted by the authors.

However, I am not entirely convinced by the pilot study. The experiment is carried out on a single task, and the proposed framework is suitable for multi-task. The discovery of single-task scenarios may not be applicable to multi-task scenarios.  A more convincing comparison is to show that "using images synthesized by an off-the-shelf generative model may hurt the performance on multi-task learning". In addition, I think using Taskonomy's output as the ground truth to train with the synthesized image is not good enough. Although Taskonomy is the state-of-the-art method, its qualitative performance is still far from being regarded as a good ground truth. It is better to visualize the sample ground truth (Taskonomy's output) in the supplement to prove that the output of using Taskonomy is sufficient for training. In general, I think the pilot study is still very different from the proposed framework: it is conducted on a single task, there is no self-supervision/contrastive learning, it does not use weak labels, but uses the output of Taskonomy as the ground truth. I think the pilot study alone is not enough to support the claim.

I also don't fully believe in the benefits of introducing generative models in joint training. The role of generative models is like providing additional data augmentations. The advantage of training the generative model with the multi-task model is that it can better learn the synthetic images guided by the multi-task outputs and the downstream tasks. I think a stronger baseline is to train a separate generative model for each task on the same dataset for each experiment, and then fix the generative model weights, train a multi-task learning network with the generated model guided by the downstream tasks. Again, since I believe that the pilot study is not enough to be such a baseline, further comparisons are needed.

**Summary Of The Paper:**

This paper focuses on using generative models to improve multi-task learning. The authors propose a framework that combines discriminative multi-task model and generative model, with a refinement network that performs scene classification on top of the multi-task network predictions to guide the learning. The network uses weak labels and self-supervision for end-to-end training. Compared with single-task and multi-task learning baselines, both quantitative and qualitative results show improved performance.

**Summary Of The Review:**

Although I think the whole idea is interesting and worthy of discussion in the research community, I still don’t fully believe most of the claims: e.g. the pilot study that tries to support the claim "directly using images synthesized by an off-the-shelf generative model (self-attention GAN) may hurt the performance on the downstream task" and the performance differences between using generative models and a more complex data augmentation technique. Therefore, at this stage, I will vote for borderline reject. I will re-evaluate the score based on the authors' response.

---

> ### Author Response · Authors · 2021-11-23
> **Response to Reviewer Ytpa 1/2**
>
> We thank the reviewer for the valuable comments. The comments focus mostly on the pilot study and joint training of the generative model and discriminative models. We address all the points as follows.
>
> 1. Pilot study is different from the proposed framework: there is no self-supervision/contrastive learning, it does not use weak labels:
>
> We thank the reviewer for the comment, and we agree with the reviewer that the pilot study is still different from our proposed MGM framework in several aspects. However, the goal of the pilot study is **not** to motivate the specific components and specific design choices proposed in our framework, such as using self-supervision and using weak labels. These components and design choices are motivated and explained in Section 3 “Method”.
>
> Instead, the pilot study aims to point out the first and foremost problem -- the difficulty of using off-the-shelf generative models for multi-task learning -- this then shows the importance of our proposed framework, which learns generative models in a way that directly facilitates multi-task learning. More concretely, as mentioned in the first paragraph of Section 2 “Pilot Study”, we show that “directly using images synthesized by an off-the-shelf generative model that is trained with the realistic objective is not helpful for downstream pixel-level perception tasks.” To sum up, we mainly identify the problem in the pilot study, and leave the solution (e.g., using self-supervision and using weak labels) in the Method Section.
>
> 2. Pilot study is carried out on a single task, and the proposed framework is suitable for multi-task:
>
> First, the pilot study shows that even for a single task, directly using images synthesized by an off-the-shelf generative model cannot benefit the downstream task, let alone for the more complicated multi-task scenario.
>
> Second, conducting a similar pilot study for the multi-task scenario would be difficult. To conduct such a pilot study, we need to generate **paired RGB and pixel-level annotations** for multiple tasks. While the RGB images can be synthesized by a GAN model, generating corresponding pixel-level annotations for the synthesized images is not trivial, as mentioned in the Paragraph “How to generate pixel-level annotations” of the Pilot Study Section. Using human annotation is expansive and time-consuming. Therefore, for semantic segmentation, we leverage an oracle annotator -- a state-of-the-art large fully supervised semantic segmentation network from Taskonomy. However, similar kinds of oracle annotators for other tasks, such as surface normal and depth estimation, are not available on Taskonomy. Therefore, due to the lack of pixel-level annotations of surface normal and depth for synthesized images, it is difficult to perform the pilot study jointly on these tasks. Moreover, this difficulty has motivated us to design a model that can leverage generative models with weak, image-level annotations to facilitate multi-task visual learning.
>
> Third, our proposed MGM framework is general and suitable for both single tasks and multiple tasks. The use of MGM for the single-task scenario is straightforward. The single task results are provided in Section G.3 and Table 15 in the Appendix.
>
> 3. Pilot study uses Taskonomy’s output as the ground truth:
>
> We respectfully disagree with the reviewer on using Taskonomy’s output as the ground truth to train with the synthesized images. This is because on Taskonomy the semantic segmentation annotations of the real images are labeled by the oracle annotator (not by humans) -- a state-of-the-art large fully supervised semantic segmentation network (please refer to the detail in Supplementary Section 14 of the Taskonomy paper). That is, consistent with the Taskonomy paper, we use the oracle annotator model’s output as the ground truth for both the real images and synthesized images, making their annotation consistent with each other.
>
> Per the reviewer’s request, we visualized the sample semantic segmentation annotations produced by the Taskonomy oracle annotator in Section I and Figure 8 in the Appendix. The visualization shows that the annotation is accurate and using Taskonomy is sufficient for training.

---

> > ### Author Response · Authors · 2021-11-23
> > **Response to Reviewer Ytpa 2/2**
> >
> > 4. Additional experiments with MGM on single tasks & stronger baseline
> >
> > We thank the reviewer for suggesting this additional baseline. To answer the reviewer’s question, we would like to first clarify that the key challenge of using generative models for pixel-level downstream tasks (**irrespective of a single task or multiple task**) is not only synthesizing useful RGB images, but also generating the corresponding pixel-level annotations for the synthesized images. While the RGB images can be synthesized by a GAN model, generating corresponding pixel-level annotations for the synthesized images is not trivial, as mentioned in the Paragraph “How to generate pixel-level annotations” of the Pilot Study Section. Our MGM framework overcomes this difficulty and enables us to use synthesized images paired with only weak annotations (i.e., image-level scene labels) to facilitate downstream tasks. This is achieved by jointly training the generative model together with the discriminative models of the downstream tasks. And this is the case irrespective of whether the downstream task is a single task or multiple tasks.
> >
> > Therefore, we would like to kindly point out that the naive version of the baseline, as suggested by the reviewer “training a separate generative model for each task and then fixing the generative model weights and training a multi-task learning network”, is not feasible. This is because, as mentioned before, even for a single task, it is difficult to train a generative model that produces paired images and pixel-level annotations.
> >
> > By contrast, our MGM framework is able to train a generative model to produce images paired with image-level weak annotations that facilitate a single downstream task. Note that the key here is still to jointly train the generative model and the single-task discriminative model. In Section G.3 and Table 15 in the Appendix, we include the results of three MGM variants, each trained for an individual task, in the NYUv2 50% data setting. Our MGM-based models consistently outperform the single task baselines ST: for semantic segmentation mIOU (larger is better) -- MGM (0.244) vs. ST (0.230); for depth estimation mABSE (smaller is better) -- MGM (0.752) vs. ST (0.837); for surface normal mAD (smaller is better) -- MGM (0.277) vs. ST (0.309). This shows that even trained on a specific task and without knowledge from relevant tasks, MGM can still benefit the individual downstream task.
> >
> > Based on this, in Section G.3 and Table 15 in the Appendix, we conducted a baseline named MGM-Combine, which is similar to what the reviewer suggested -- we first train the three separate MGM variants, each trained for an individual task; we then fix the generative model weights and train a multi-task learning network. Comparing the performance between MGM-Combine (SS 0.249 mIOU; DE 0.747 mABSE; SN 0.277 mAD) and MGM (SS 0.251 mIOU; DE 0.734 mABSE; SM 0.273 mAD), we can find that after we freeze the generative models, the performance is worse than the full MGM model. This result indicates that (1) learning from a single task can improve the performance for that target task; (2) but the joint training mechanism can further benefit all the downstream tasks, by sharing the knowledge across different tasks.

---

### Official Review · Reviewer_zrdP · 2021-11-03

**Correctness:** 4
**Technical Novelty And Significance:** 3
**Empirical Novelty And Significance:** 3
**Recommendation:** 6
**Confidence:** 4

**Main Review:**

Pros)
The idea on introducing an image generation network, G,  into multi-task learning on pixel-wise estimation tasks seems novel. To use scene-class-conditionally synthesized images without GT annotations of the target multi-tasks, the authors introduced a refinement network which classifies a scene class. The scene class was known for a synthesized image, since it was generated with a class condition, which enabled training of the multi-task network,M, with generated images.  In addition, EM-style training was proposed to train the network effectively. Its effectiveness was successfully proved by the experiments. This is the biggest contribution of this paper.

Although adding a self-supervision network is not so novel, its effectiveness was shown by the experiments.

The paper is well written, and the experiments including ones in the supplementary material are comprehensive and detailed.
The pilot study is helpful to understand that naive usage of GAN for this task is not effective at all.

Cons)
The proposed method needs to scene labels for all the training images. That's why MGM cannot be applied to the CityScape dataset which consists of only street scene. This property narrows applicability of the proposed method.

Additional Comments)
To deal with CityScape, it will be needed to introduce a multi-label classifier into the refinement network, and use a multi-label-conditional
GAN (probably using multiple-hot instead of one-hot vector). Does the authors think if this works ? In Semi Supervised Semantic Segmentation Using Generative Adversarial Network,ICCV2017, multi-label was used.

MGM does not seem to be limited to multiple tasks. It can be applied to a single task such as only semantic segmentation. Did the authors try that ?

Showing the synthesized images on the supplementary material will be helpful for the readers.


**Summary Of The Paper:**

This paper proposes a multi-task network for pixel-wise estimation tasks such as semantic segmentation and depth estimation. The authors additionally introduced an GAN-based image generation network, a self-supervised network and a scene category classification network (which was called a refinement network in the paper). As an image generation network, any conditional GANs including SA-GAN and DCGAN can be used. As a self-supervised network, SimCLR was used in the proposed network. The refinement network classifies
a scene category from the outputs of multitask networks (i.e. estimated segmentation masks and estimated depth images).

In the experiments with two dataset, the proposed model, MGM, consistently outperformed the results of the baselines and ablated methods.
In addition, additional experiments with six tasks and high resolution images (256x256) were performed and confirmed the effectiveness of the proposed method as well.

**Summary Of The Review:**

This paper has novelty which is introducing a generative model into multitask network, and the effectiveness was successfully supported by the comprehensive experiments.  The reviewer think this paper can be accepted.

---

> ### Author Response · Authors · 2021-11-23
> **Response to Reviewer zrdP**
>
> We thank the reviewer for the positive comments and the recognition of our work. The comments focus mostly on the applicability of our approach in different scenarios and the visualization of synthesized images. We address all the points as follows.
>
> 1. Can MGM be applied to datasets like CityScape, where there is no, or only a single type of, scene label but multi-hot object labels:
>
> We thank the reviewer for the suggestion. Our method is applicable to datasets like CityScape, where either scene labels are not available or there is only a single type of scene (e.g., CityScape consists of only a street scene). This can be achieved in a way that, as suggested by the reviewer, replaces the one-hot scene labels by the multi-hot object labels. And this is because our generative network and refinement network can naturally work with multi-hot labels -- the refinement network can work with a multi-label classifier, and the generative network can be a multi-label-conditional GAN.
>
> Following the reviewer’s suggestion, we conducted an experiment on a subset of the CityScape dataset, the Zurich street scene. We focused on the semantic segmentation task, which is a representative task on CityScape. The detailed experimental setting is described in Section G.1 in the Appendix, and the experimental result is shown in Table 13. MGM (0.64 mIOU) still outperforms the single task baseline ST (0.57 mIOU), indicating the robustness and generalizability of our model with multi-hot object labels.
>
> 2. Does MGM work for single tasks:
>
> MGM is a general framework that can be applied to both single tasks and multiple tasks. In the original submission, we mainly focused on the more challenging multi-task scenario. The use of MGM for the single-task scenario is straightforward. Note that, though, in the single-task scenario, the model cannot leverage useful information from multiple relevant tasks.
>
> In Section G.3 and Table 15 in the Appendix, we include the results of three MGM variants, each trained for an individual task, in the NYUv2 50% data setting. Our MGM-based models consistently outperform the single task baselines ST: for semantic segmentation mIOU (larger is better) -- MGM (0.244) vs. ST (0.230); for depth estimation mABSE (smaller is better) -- MGM (0.752) vs. ST (0.837); for surface normal mAD (smaller is better) -- MGM (0.277) vs. ST (0.309). These results show that even trained on a specific task and without shared knowledge from multiple relevant tasks, MGM can still benefit the individual downstream task.
>
> 3. Visualization of synthesized images:
>
> We thank the reviewer for the suggestion. This paper targets the performance of multi-task learning, therefore the visual quality of the synthesized images is not our main focus, and we did not visualize them in the original submission.
>
> Following the reviewer’s suggestion, we visualized the synthesized images for both off-the-shelf SAGAN and our MGM with joint training in Section H and Figure 7 in the Appendix. We have the following observations:
>
>   - The conventional SAGAN, trained with the realism objective and without the guidance of the downstream tasks, produces photo-realistic images. However, as the pilot study shows, these photo-realistic images do not facilitate downstream tasks.
>   - The visual quality of the images synthesized by MGM becomes degraded. Here, MGM trains a SAGAN-based generative model jointly with the multi-task learning objective and under the guidance of the downstream tasks. While these synthesized images are not photo-realistic, they do improve the performance of the downstream tasks.
>
> Hence, there seems to be an interesting trade-off between photo-realism and usefulness for improving downstream discriminative tasks. A similar phenomenon has been observed in [Ref1], where a generative model is trained to facilitate the semantic segmentation task.
>
> We hypothesize that this is because those synthesized images that are useful for improving downstream tasks might not be necessarily photo-realistic. From Figure 7, we can see that, although not visually very realistic, the images synthesized by MGM still contain patterns related to objects in the scene and encode spatial object distributions and scene layout. These images thus increase the diversity of the training data and are useful for training the discriminative models of the downstream tasks.
>
> [Ref1] Nasim Souly, Concetto Spampinato, and Mubarak Shah. “Semi supervised semantic segmentation using generative adversarial network.” In ICCV, 2017.

---

### Author Response · Authors · 2021-11-23
**General Response to All the Reviewers**

We thank all reviewers for their interest in our approach and their constructive and valuable comments.

We have uploaded a new version of the submission that takes into account all the comments from the reviewers. The revised text is marked as blue.

The main revision is summarized as follows:

  1. We revised the text on the analysis of the main results in Section 4.2, to better clarify our observations in the 25% data setting.
  2. We included additional discussion with related work suggested by Reviewer zrdP in the paragraph “Generative Modeling for Visual Learning” in Section 5.
  3. We added additional experimental evaluations in Section G in the Appendix, including
      - MGM with multi-hot object labels on a sub-dataset of CityScape;
      - MGM in an even lower data regime on the Tiny-Taskonomy dataset;
      - MGM for single tasks;
      - An additional baseline suggested by Reviewer Ytpa.
  4. We added and compared visualization of the synthesized images by off-the-shelf SAGAN and MGM in Section H in the Appendix.
  5. We visualized the semantic annotations which we used in the pilot study in Section I in the Appendix.
  6. We revised Figure 6 in the Appendix to correct the mismatch of the visualization and also increased the image resolution for better visualization.

---

### Decision · Program_Chairs · 2022-01-20

**Decision:**

Reject

**Comment:**

The paper received borderline reviews. While the reviewers acknowledged good motivation, good number of experiments and good numeral results that demonstrated the proposed method outperforms the existing state of the art, there are shared concerns: the experimental setup is not really a "low data" regime, generative models jointly trained with the multi-task model only led to marginal improvements, and the prediction quality is quite low for all methods. In addition, it's unclear why the images generated by MGM have a lot of artifacts, and how the artifacts affect the performance. Overall, the reviewers were not convinced after the rebuttal.